# In situ-formed tetrahedrally coordinated double-helical metal complexes for improved coordination-activated n-doping

Ziyang Liu[1], Xiao Li[1], Yang Lu[2], Chen Zhang[1], Yuewei Zhang[1,3], Tianyu Huang[1], Dongdong Zhang [1,3✉] & Lian Duan [1,3✉]

In situ coordination-activated n-doping by air-stable metals in electron-transport organic ligands has proven to be a viable method to achieve Ohmic electron injection for organic optoelectronics. However, the mutual exclusion of ligands with high nucleophilic quality and strong electron affinity limits the injection efficiency. Here, we propose meta-linkage diphenanthroline-type ligands, which not only possess high electron affinity and good electron transport ability but also favour the formation of tetrahedrally coordinated double-helical metal complexes to decrease the ionization energy of air-stable metals. An electron injection layer (EIL) compatible with various cathodes and electron transport materials is developed with silver as an n-dopant, and the injection efficiency outperforms conventional EILs such as lithium compounds. A deep-blue organic light-emitting diode with an optimized EIL achieves a high current efficiency calibrated by the y colour coordinate (0.045) of 237 cd A$^{-1}$ and a superb LT95 of 104.1 h at 5000 cd m$^{-2}$.

[1] Key Lab of Organic Optoelectronics and Molecular Engineering of Ministry of Education, Department of Chemistry, Tsinghua University, Beijing 100084, China. [2] Institute of Drug Discovery Technology, Ningbo University, Ningbo 315211, China. [3] Center for Flexible Electronics Technology, Tsinghua University, Beijing 100084, China. ✉email: ddzhang@mail.tsinghua.edu.cn; duanl@mail.tsinghua.edu.cn

Charge injection at metal/organic interfaces plays a vital role in improving the efficiencies and stabilities of organic semiconductor electronics[1–6]. Among all endeavors to modulate the energy level alignment and improve the charge injection efficiency, n-doping techniques that not only allow tuning of the work function over a wide range but also promote electrical conductivity of the semiconductor layers have attracted considerable attention[7]. Specifically, the ionization energy (IE) of n-dopants is lower than the electron affinity (EA) of organic semiconductors as electron transport materials (ETMs), thus making spontaneous charge transfer from n-dopants to the lowest unoccupied molecular orbitals (LUMOs) of ETMs energetically favorable. As a result, the Fermi level ($E_F$) shifts towards the LUMOs of the ETMs, enhancing electron injection[8]. Species with sufficiently low IEs, including reactive alkali metals and a few easily oxidized molecular compounds, are thus necessary, particularly for commercialized organic light-emitting diodes (OLEDs), as ETMs always have a small electron affinity (usually <3.0 eV)[9]. Despite their hazardous reactivity and diffusivity, highly reactive metals are still widely employed today to facilitate efficient n-doping and ensure desirable device performance towards commercial application[10–12]. Although air-stable alkali salt precursors (e.g., $Cs_2CO_3$[13] and $Rb_2CO_3$[14]) that can liberate alkali metals during the fabrication process have been exploited to address this issue, the resulting n-doping process always suffers from undesired outgassing and metal diffusion. Air-stable, vacuum-deposited, byproduct-free and widely applicable n-doping strategies remain vigorously ongoing pursuit until now[15,16]. To achieve this goal, Kahn et al. developed a cleavable air-stable organometallic dimer as an n-dopant precursor, the photoactivated cleavage of which yields two monomers for efficient n-doping of materials with low EAs[17]. Fukagawa et al also demonstrated that by adding a superbase at the ETM/cathode interface, the work function (WF) near the aluminum (Al) cathode can be tuned to ~2.0 eV through a coordination reaction as well as the formation of H-bonds[18,19]. The preliminary work revealed the viability of replacing alkali metals with conceptual advancements in n-doping techniques.

Recently, our group proposed that in the presence of chelating ligands, air-stable metals, such as copper (Cu), silver (Ag), and gold (Au), could readily release free electrons, as the irreversible coordination reaction between metal cations and the ligands would shift the equilibrium between the metal atoms and metal cations in the forward direction[20,21]. This in situ coordination-activated n-doping (CAN) strategy allows the WFs of air-stable metals such as Ag to be feasibly modulated by manipulating the chelating ligands. Featuring rigid heterocyclic planar structures with preorganized coordinating nitrogen sites, 1,10-phenanthroline (Phen) derivatives have exhibited good chelating ability with superior electron transport properties[22–24]. By substituting the 4,7-position of the Phen ring with strong electron-donating groups such as methoxyl and pyrrolidin-1-yl groups, the nucleophilicity of the N-chelating sites can be greatly enhanced, rendering Ag an even stronger n-dopant than cesium (Cs)[21]. The CAN strategy shows the potential to revolutionize the n-doping technique of the OLED industry, as it is byproduct-free and highly effective. Nevertheless, for efficient charge injection layers, not only feasible charge injection but also good charge transport is necessary. In the CAN system, chelating ligands also function as electron acceptors and transporters, satisfying the need for both high nucleophilicity and transport ability. The most commonly adopted tactic to enhance the chelating ability of ligands by introducing donor groups seems to be a double-edged sword since the electron affinity and electron transport ability of ligands will be significantly reduced. Thus, the charge injection efficiency is limited, as reflected by the strong dependence on the

concentration of Ag and the thickness of the electron injection layer (EIL)[21,25]. Therefore, determining how to eliminate this intractable trade-off between nucleophilic ability and electron affinity when developing ligands is urgently needed if a widespread transition towards practical application in the OLED industry is to occur.

Here, we introduce a conceptual advancement in the development of ligands for CAN, that is, meta-linked diphenanthroline compounds as electron-transport ligands (ET ligands). These ligands were found to feasibly form tetrahedrally coordinated double-helical metal complexes during the co-evaporation process, effectively lowering the IE of air-stable metals to achieve improved electron injection. Benefiting from the high charge transport ability of ligands, efficient electron injection independent of the thickness of the EILs and the dopant concentration over a large range was realized. By considering the chelating dynamics, a meta-linked diphenanthroline ligand (m-dPhen) with a reduced steric effect was elaborately developed, achieving better charge injection than the commonly used lithium fluoride (LiF) and 8-hydroxyquinolinato lithium (Liq) as well as a high maximum external quantum efficiency (EQE) of 10.3% for deep-blue devices based on triplet-triplet annihilation (TTA), which is among the highest EQE values for TTA-driven OLEDs with y color coordinates below 0.1. Moreover, the device performance of the deep-blue OLED with a Commission Internationale de l'Eclairage (CIE) coordinate of (0.139 and 0.045) satisfying the requirement of BT.2020 is further improved by implementing m-dPhen as the EIL, thus resulting in a high current efficiency calibrated by the y color coordinate of 237 cd A$^{-1}$ at 1000 cd m$^{-2}$ and a superlong LT95 (time to 95% of the initial luminance) of 104.1 h at an initial luminance of 5000 cd m$^{-2}$. This performance is better than that achieved with commercial EILs of ytterbium (Yb), thus validating the potential of CAN as an alternative n-doping technique applicable for efficient optoelectronics.

## Results

**Effects of chelating ligands on the coordination reaction.** As a proof-of-concept ET ligand, 1,3-bis(9-phenyl-1,10-phenanthrolin-2-yl)benzene (m-dPPhen) was first studied. For reference, 1,4-bis(9-phenyl-1,10-phenanthrolin-2-yl)benzene (p-dPPhen) was also developed (Supplementary Fig. 1). Figure 1a shows the single-crystal structures of the two ligands, revealing the formation of multiple N···H−C hydrogen bonds for both ligands due to the nitrogen-containing heterocycles. Interestingly, owing to the strong hydrogen bonds with short distances of 2.468 and 2.487 Å, small dihedral angles of 8.83° and 9.03° between the 2-phenyl-1,10-phenanthroline (2PPhen) units and the central phenyl ring in m-dPPhen were obtained. In contrast, relatively large dihedral angles of 22.41° and 22.95° were observed for p-dPPhen, which were attributed to its relatively weak hydrogen bonds with longer H-bond distances of 2.520 and 2.528 Å, respectively.

To study the chelating behaviors between ligands and metals, the two ligands were separately mixed with equal amounts of silver nitrate (AgNO$_3$) in an acetonitrile/methanol (CH$_3$CN/CH$_3$OH) solution. After removing the solvent, a single crystal was obtained for the reaction products of m-dPPhen. A double-helical metal complex structure of [Ag$_2$(m-dPPhen)$_2$](NO$_3$)$_2$ was confirmed by X-ray diffraction analysis of the single crystals. As shown in Fig. 1b, each silver atom is ligated by two Pphen groups due to the strong coordination between the nitrogen and silver atoms. The distances between the silver and adjacent nitrogen atoms range from 2.3 to 2.4 Å. Although no crystal was obtained for the mixture of p-dPPhen and AgNO$_3$ with a molar ratio of 1:1, the different chelating behaviors were confirmed by $^1$H nuclear magnetic resonance (NMR) measurements, and the results of pristine

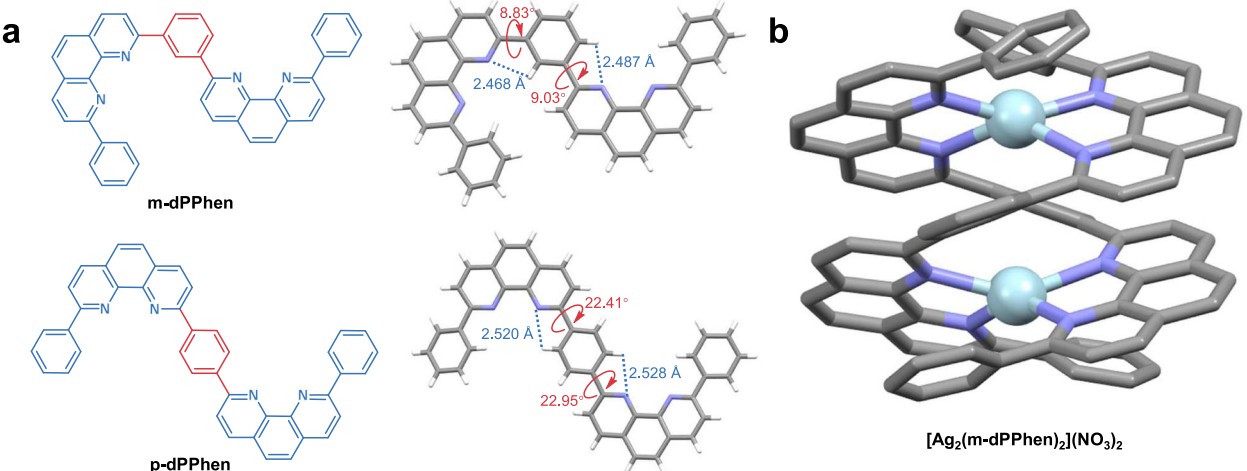

**Fig. 1 The molecular structures of different diphenanthroline ligands and double-helical metal complexes. a** Molecular structures and single-crystal structures of p-dPPhen and m-dPPhen. **b** Single-crystal structures of [Ag$_2$(m-dPPhen)$_2$](NO$_3$)$_2$.

**Fig. 2 The coordination reaction of different diphenanthroline ligands. a** $^1$H NMR spectra of the resulting products of p-dPPhen:AgNO$_3$ (molar ratio of 1:1) in CDCl$_3$. **b** $^1$H NMR spectra of the resulting products of m-dPPhen and AgNO$_3$ (molar ratio of 1:1) in CDCl$_3$. **c** Electrostatic potential maps of monomolecular m-dPPhen. **d** Electrostatic potential maps of two m-dPPhen in a double-helical structure.

diphenanthroline ligands were used for comparison (Fig. 2a, b)[19]. In particular, a shielding effect induced by the 2PPhen cores was observed from the products obtained from m-dPPhen and AgNO$_3$. Compared to the free ligands (m-dPPhen) with all $^1$H peaks in the range of 7.4–9.8 ppm, the coordination of m-dPPhen and AgNO$_3$ caused the shift of all $^1$H peaks to a higher magnetic field, which has been regarded as a typical feature of double-helical

structures[26,27]. However, the major $^1$H peaks of p-dPPhen:AgNO$_3$ were almost identical to those of pristine p-dPPhen, in accordance with previously reported results for the Bphen:AgNO$_3$ system[20]. These results imply weak coordination bonds between p-dPPhen and Ag$^+$, and thus, only [Ag(p-dPPhen)](NO$_3$) is expected to be the dominant product. Bearing the same 2PPhen moieties, the different coordination behaviors of p-dPPhen and m-dPPhen can

be attributed to the different configuration effects of the π-linkages of these ligands.

To evaluate the chelating ability of these two ligands, electrostatic potential (ESP) maps were obtained by density functional theory (DFT) calculations with the B3LYP/6-31 G(d) basis set. The maximum ESPs of both molecules were found to be mainly distributed around the nitrogen atoms of the 2PPhen ligands, indicating that nitrogen atoms are the active binding sites for the coordination reaction (Supplementary Fig. 2). Maximum ESPs of −0.059 and −0.061 were observed for p-dPPhen and m-dPPhen, respectively, which are lower than those of mono-Phen derivatives obtained in our previous work[18]. We further calculated the ESP of the double-helical structure of the m-dPPhen dimer, and a high value of −0.140 was obtained, validating the high nucleophilicity of the N atoms in this structure (Fig. 2d). These ESP values suggest that this unique m-dPPhen dimer can better reduce the IEs of air-stable metals than mono m-dPPhen.

In terms of the electron affinity, cyclic voltammetry (CV) was adopted to identify the highest occupied molecular orbital (HOMO) and lowest unoccupied molecular orbital (LUMO) energy levels of target molecules combined with optical energy gaps ($E_g$s) (Supplementary Figs. 3, 4). Similar reversible reduction processes were observed at negative bias, indicating their potential as stable electron transport materials. The HOMO/LUMO energies of p-dPPhen and m-dPPhen identified by CV measurements are −6.0/−3.1 eV and −6.0/−3.0 eV, respectively. Compared with the mono-Phen ligands with electron-donating substituents at the 4,7-positions, the LUMO levels of which are as high as −2.4 eV[21], diphenanthroline derivatives exhibited lower LUMO energy levels, indicating their high electron affinity as anticipated. In addition, mono-Phen ligands, especially those with electron-donating substituents at the 4,7-positions, suffer low electron mobility ($μ_e$), which will hinder electron hopping between adjacent Phen planes. In contrast, diphenanthroline-type ETMs have demonstrated $μ_e$ values nearly one order of magnitude higher than that of Bphen[24]. Moreover, on account of their rigid planar structures and high molecular weights (Supplementary Figs. 5−7), the two molecules show good thermal stability. Differential scanning calorimetry (DSC) and thermo-gravimetric analysis (TGA) were applied to the two molecules. The designed p-dPPhen and m-dPPhen were found to have high glass-transition temperatures ($T_g$) of 150/138 °C and decomposition temperatures ($T_d$, corresponding to 5% weight loss) of 469/451 °C, respectively. Considering that mono-Phen ligands usually suffer from poor thermal stability[28], the high $T_g$s of diphenanthroline derivatives should improve the thermal and operational stability of the doped films by CAN strategies (Supplementary Fig. 8 and Supplementary Table 1).

All the results above suggest that m-dPPhen exhibits the potential to break the mutual exclusion of high nucleophilicity and electron affinity in mono-Phen derivatives. To further evaluate the performance of dPhen-type ligands in CAN, electron-only devices (EODs) with structures of ITO/4,7-diphenyl-1,10-phenanthroline (Bphen):Cs$_2$CO$_3$ (10 nm, 10 wt%)/9,10-bis(6-phenylpyridin-3-yl) anthracene (DPPyA, 100 nm)/dPhen-type ligands:Ag (5 nm)/Al (150 nm) were fabricated. With pristine dPhen ligands as a reference, the Ag doping concentrations were manipulated to be 5, 10, and 20 wt%. With the same device configurations, the current density ($J$)−voltage ($V$) characteristics of EODs depend exclusively on the performance of the EILs. As shown in Fig. 3a, the current densities of EODs with Ag-doped EILs are obviously enhanced in comparison to those involving pristine dPhen ligands as EILs, confirming the importance of Ag as an n-dopant. In addition, the current densities of EODs with m-dPPhen:Ag as EILs are obviously

higher than those involving EILs of p-dPPhen:Ag at the same doping concentrations. Rather low driving voltages of 1.3 and 2.4 V were recorded to attain a high current density of 100 mA cm$^{-2}$ for EODs based on m-dPPhen:Ag (5 nm, 10 wt%) and p-dPPhen:Ag (5 nm, 10 wt%). To quantify the electron injection properties of EILs, the electron injection efficiency ($η_{injection}$) at the optimized doping concentration of 10 wt% was expressed by the current density −average applied electric field ($J_{EOD}$−$E$) characteristics of EOD and theoretical space-charge limited current-applied electric field ($J_{SCLC}$−$E$) curves as follows[29]:

$$η_{injection} = \frac{J_{EOD}}{J_{SCLC}} \quad (1)$$

Based on the assumption of barrier-free (Ohmic) injection, the injected current densities of EODs should follow the space-charge limited current (SCLC) and can be calculated by Eq. (2):

$$J_{SCLC} = \frac{8}{9}μ_0ε_0ε_r\exp\left(β\sqrt{E}\right)\frac{E^2}{L} \quad (2)$$

Where $ε_o$ is the vacuum permittivity and $ε_r$ is the relative permittivity, which is assumed to be three for organic thin films. $E$ represents the average applied electric field and $L$ is the thickness of the organic semiconductor layers. $μ_0$ stands for the zero-field mobility, and $β$ represents the Poole–Frenkel slope of electron transport materials. Herein, $μ_0$ of $4.6 × 10^{-4}$ cm$^2$ V$^{-1}$ s$^{-1}$ and $β$ of $1.56 × 10^{-3}$ cm$^{-1/2}$ V$^{1/2}$ were determined by the time-of-flight (TOF) transient photocurrent technique according to previous reports[30]. Accordingly, the electron injection efficiency ($η_{injection}$)−electric field ($E$) curves of optimized EODs based on m-dPPhen and p-dPPhen are plotted in Fig. 3b. The EILs with m-dPPhen:Ag show high $η_{injection}$ values in the range of 30–40% when the electric field varies from 100 to 220 kV cm$^{-1}$, which are approximately three times higher than those of p-dPPhen:Ag.

In general, the injected current densities of the EOD are sensitive to the electron injection barrier, which is governed by the work function of the electrode and the EA of the electron transport materials. Considering the LUMO level of DPPyA (−3.0 eV) and the conversion factors between different measurement sources[31–33], the EA of DPPyA was estimated to be smaller than 3.0 eV. A CAN-modified Al cathode with a low work function would be beneficial to achieve high injection efficiency because of the small electron injection barrier. To explain the different injection behaviors in terms of the energy level alignment, ultraviolet photoelectron spectroscopy (UPS) measurements were conducted to verify their roles in cathode WF tunability. As shown in Fig. 3c, upon depositing EILs of Ag-doped diphenanthroline ligands (5 nm, 10 wt%), the WF of the Al cathode was reduced from 3.6 to ~3.0 eV, which is very close to the LUMO levels of common ETMs, thus favoring efficient electron injection. In contrast, the p-dPPhen:Ag (5 nm, 10 wt %)-modified Al showed only a moderate WF value of 3.1 eV, explaining its inferior injection efficiency.

The differences in the injection performance were ascribed to the different chelating behaviors of the two ligands. To unravel this result, the species in Ag-doped diphenanthroline films (130 nm, 10 wt%) were investigated by MALDI-TOF measurements. As plotted in Fig. 3d, strong mass peaks at 693.13 and 1279.13 M were observed for m-dPPhen, corresponding to organometallic complexes of [Ag(m-dPPhen)]$^+$ and [Ag(m-dPPhen)$_2$]$^+$, respectively. In contrast, for the films based on p-dPPhen:Ag, strong mass peaks at 586.2 and 693.26 M were attributed to the p-dPPhen and [Ag(p-dPPhen)]$^+$ monomers. Considering their comparable ESPs and steric hindrance for

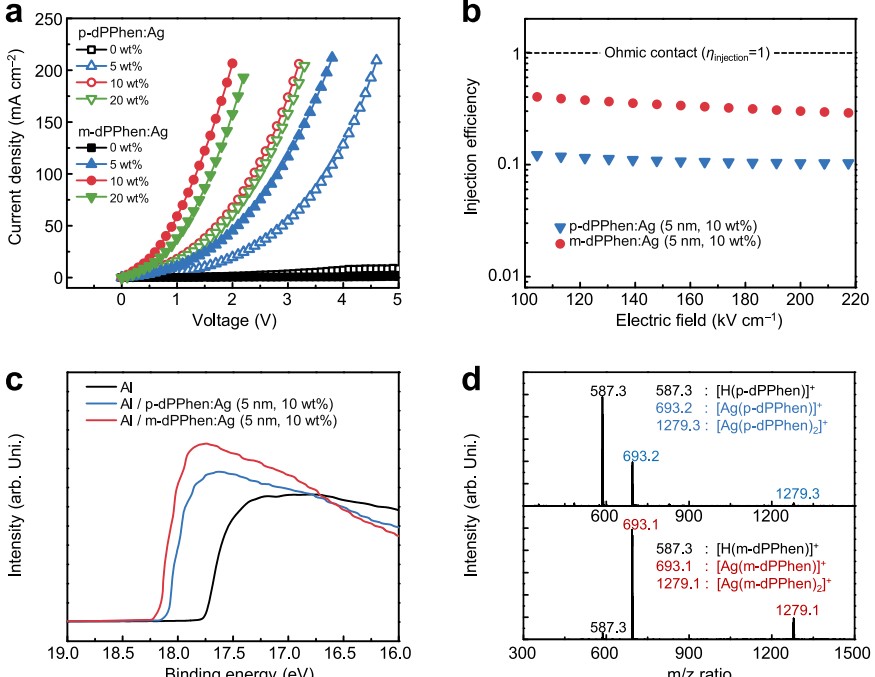

**Fig. 3 Characteristics of EILs based on Ag-doped p-dPPhen or m-dPPhen. a** The current density ($J$)−voltage ($V$) characteristics of EODs with structures of ITO/Bphen:Cs$_2$CO$_3$ (10 nm, 10 wt%)/DPPyA (100 nm)/p-dPPhen:Ag or m-dPPhen:Ag (5 nm, 0, 5, 10, 20 wt%)/Al (150 nm). **b** The injection efficiencies of EILs consisting of p-dPPhen:Ag (5 nm, 10 wt%) and m-dPPhen:Ag (5 nm, 10 wt%). **c** UPS analysis of Al (10 nm), Al (10 nm)/p-dPPhen:Ag (5 nm, 10 wt%) and Al (10 nm)/m-dPPhen:Ag (5 nm, 10 wt%). **d** Mass spectra of Ag-doped diphenanthroline films of p-dPPhen:Ag (130 nm, 10 wt%) and m-dPPhen:Ag (130 nm, 10 wt%). The molar ratio of diphenanthroline ligand:Ag is 1.84:1, and the doping concentration is 10 wt%.

monomers, their different coordination behaviors were assigned to the fact that the meta-type linkage in m-dPPhen is conducive to the formation of tetrahedrally coordinated double-helical species. As mentioned above, the double-helical structure of m-dPPhen greatly increases the ESP values and thus favors lowering the WF of the Al cathode to realize improved injection.

**Optimization of diphenanthroline ligands for CAN.** Previous studies have validated that the phenyl ring attached to the 2-position of Phen usually introduces steric effects and lowers the nucleophilic quality of ligands[20]. A more advanced ligand, 1,3-di(1,10-phenanthrolin-2-yl)benzene (m-dPhen), was thereby constructed to eliminate the influence of steric effects. The X-ray diffraction analysis of m-dPhen suggested that intramolecular N···H−C hydrogen bonds in m-dPhen between the phenyl groups and Phen groups led to the U-shaped molecular conformation in the single crystals (Fig. 4a and Supplementary Fig. 6). This unique U-shaped structure results in a synergetic effect of the Phen units and leads to an increased maximum ESP of −0.087 compared with that of m-dPPhen (−0.061). The increased ESP and reduced steric hindrance make coordination reactions more thermodynamically favorable. More importantly, the conformation of m-dPhen in single crystals is similar to that in double-helical compounds, suggesting that m-dPhen may undergo smaller conformational changes than m-dPPhen during coordination reactions (Supplementary Data 1). Based on these results, we calculated the IEs of in situ-formed n-dopants consisting of silver and diphenanthroline ligands by estimating the enthalpy changes during the process of Ag(ligand)$_n$ → [Ag(ligand)$_n$]$^+$ + e$^-$ (Fig. 4b). Interestingly, Ag(m-dPhen) was found to possess the lowest calculated IE of 3.47 eV among all organometallic complexes with one dPhen ligand. This finding was explained by the unique U-shaped structure of m-dPhen, which induces a synergetic effect between the two Phen units to increase the ESP. The

IEs of tetrahedrally coordinated metal complexes (AgL$_2$) were also calculated to be 3.11, 3.10 and 3.05 eV for p-dPPhen, m-dPPhen, and m-dPhen, respectively. Among these organometallic complexes, Ag(m-dPhen)$_2$ with a tetrahedrally coordinated double-helical structure showed significantly reduced IE, favoring more efficient CAN. These results reflect the validity of the electronic effect and steric effect obtained by replacing 2PPhen moieties with Phen moieties.

To investigate the coordination reaction between Ag$^+$ and m-dPhen ligands, $^1$H NMR measurements were employed to identify the coordination sites of the organometallic complex (Supplementary Fig. 9a). The downward chemical movement of the $^1$H peak suggests the formation of tetrahedrally coordinated double-helical species[21]. Additionally, mass spectrometry was also employed to study the components of m-dPhen:Ag (130 nm, 10 wt%, the molar ratio of m-dPhen:Ag = 2.48:1). The strong peaks at 541.09 and 975.21 M suggest the coordination products of [Ag(m-dPhen)]$^+$ and [Ag(m-dPhen)$_2$]$^+$, demonstrating the formation of organometallic complexes with low IEs during co-evaporation, which have been regarded as the key components for efficient n-doping (Supplementary Fig. 9b). EODs employing different EILs, including Ag-doped diphenanthroline ligands and conventional EILs, such as LiF[34] and Liq[35], with a structure of ITO/Bphen:Cs$_2$CO$_3$ (10 nm, 20 wt%)/DPPyA (100 nm)/EIL/Al (150 nm) were examined. To reasonably evaluate the electron injection properties of different EILs, we further measured the $J$−$V$ characteristics of EODs with various EILs at different thicknesses to verify their optimal thicknesses (Supplementary Fig. 11), thus determining their maximum electron injection efficiencies in the EODs. Accordingly, lithium compounds with a thickness of 1 nm and Ag-doped diphenanthroline ligands with a thickness of 5 nm were employed for comparison. The $J$−$V$ characteristics of EODs employing different EILs with optimal thickness are summarized in Fig. 4c, with m-dPhen:Ag (5 nm,

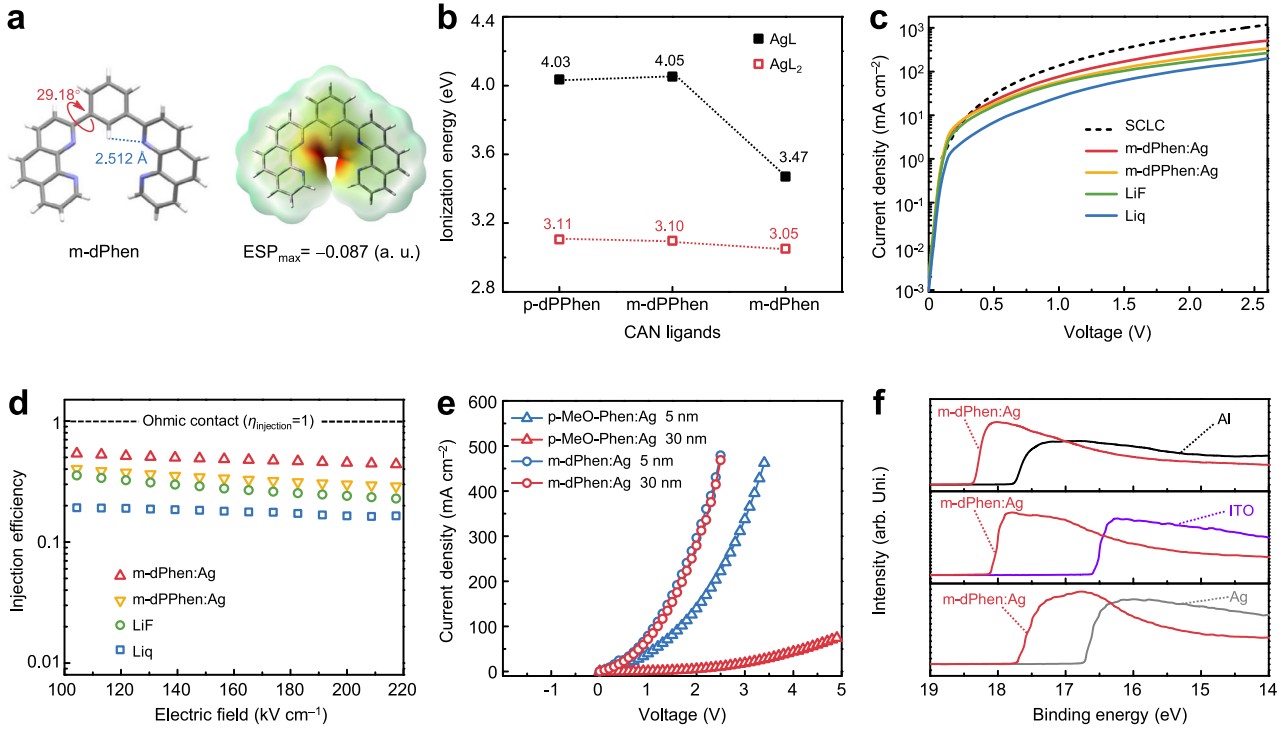

**Fig. 4 Characterization of the electron injection properties of m-dPhen. a** Single-crystal structures and ESP maps of m-dPhen. **b** Calculated ionization energies of different organometallic complexes consisting of diphenanthroline ligands and silver. **c** The current density ($J$)—voltage ($V$) characteristics of EODs based on different EILs of LiF (1 nm), Liq (1 nm), m-dPPhen:Ag (5 nm, 10 wt%), and m-dPhen (5 nm, 10 wt%). **d** The injection efficiencies of EODs based on different EILs of LiF (1 nm), Liq (1 nm), m-dPPhen:Ag (5 nm, 10 wt%), and m-dPhen (5 nm, 10 wt%). **e** The $J$—$V$ characteristics of EODs based on EILs of p-MeO-phen:Ag (5 or 30 nm, 10 wt%) and m-dPhen:Ag (5 or 30 nm, 10 wt%). **f** UPS analysis of different electrodes, including Al (10 nm), ITO (135 nm) and Ag (10 nm), with and without m-dPhen:Ag (5 nm, 10 wt%).

10 wt%) being the best. At a high current density of 100 mA cm$^{-2}$, the driving voltage of the EOD using m-dPhen:Ag was as low as 1.1 V, which is lower than those based on m-dPPhen:Ag (1.3 V), LiF (1.5 V) and Liq (1.9 V). Removal of the peripheral phenyl group at the 2-position of 2PPhen is a fruitful way to enhance the coordination reaction of diphenanthroline ligands. Figure 4d illustrates the calculated electron injection efficiencies of EODs. The m-dPhen:Ag (5 nm, 10 wt%) exhibits the highest $\eta_{injection}$ of ~50%, which is higher than that of commonly used LiF (1 nm) and Liq (1 nm) with $\eta_{injection}$ values of no more than 40% at 150 kV cm$^{-1}$.

The influence of the Ag doping concentration on the injection performance was further studied. As shown in Supplementary Fig. 12b, the $\eta_{injection}$ of m-dPhen:Ag EIL at 100 kV cm$^{-1}$ increases at first and then saturates at higher doping concentrations, which can be ascribed to the electrical n-type doping characters. Intriguingly, $\eta_{injection}$ increased by two orders of magnitude from 0.24% for neat m-dPhen to 46% with 5 wt% Ag doping. The high electron injection efficiency at a low doping concentration of 5 wt% (0.5 vol%) can also reflect the strong coordination ability of m-dPhen, which is preferred in terms of optical transparency and fabrication robustness. Moreover, as presented in Fig. 4e, the influence of the EIL thickness on electron injection ability was also investigated by using a 4,7-substituted mono-Phen (4,7-dimethoxy-1,10-phenanthroline, p-MeO-Phen) ligand as a reference. Interestingly, almost the same current densities were observed for the EODs with 5 and 30 nm m-dPhen:Ag films (10 wt%). In contrast, for the EODs with p-MeO-phen:Ag (10 wt%), the current densities sharply decreased when the EIL thickness increased from 5 to 30 nm. The current densities of the EODs with p-MeO-phen:Ag were lower than

those of the EODs with m-dPhen:Ag at the same EIL thickness of 5 or 30 nm. The different thickness dependences of Ag-doped EILs were attributed to the electron transport abilities of the ligands. As mentioned above, for the p-MeO-Phen ligand, although methoxy groups at the 4,7-position can enhance the nucleophilic quality of the ligand, both EA and $\mu_e$ were found to be reduced. By decoupling the nucleophilic quality and electron affinity, the meta-linked diphenanthroline-type ligand m-dPhen outperformed the typical mono-Phen ligand p-MeO-Phen in terms of electron injection. As a result of the strong chelating ability and high electron transport properties of diphenanthroline ligands, the m-dPhen-based CAN system retained a high electron injection efficiency within a wide processing window of thicknesses (between 5 and 30 nm) and doping concentrations (between 5 wt% and 20 wt%), thus providing good manufacturing feasibility. Since the LUMO levels of the prevailing electron transport materials in organic solar cells (OSCs) mostly range from $-4$ to $-3$ eV[9], the CAN-modified electrodes with a low work function of approximately 3.0 eV can also facilitate electron extraction in view of energy alignment. In addition, a 2,9-dimethyl-4,7-diphenyl-1,10-phenanthroline (BCP):Ag complex has been successfully employed to reduce the electron extraction barrier between a $C_{60}$ electron transport layer and indium-zinc oxide top electrode in high-performance perovskite solar cells[36]. Thus, the CAN technique possessing high WF tunability, good thermal stability, and a wide processing window is anticipated to be applicable to cathode modification in OLEDs, OSCs, and other optoelectronic devices.

UPS measurements were further conducted to explain the performance of m-dPhen in the CAN system, and it was found that the work function of the Al cathode could even be reduced to 2.8 eV with an EIL of m-dPhen:Ag (5 nm, 10 wt%), energetically

favoring electron injection because of the energy level alignment. Moreover, it was found that a 5-nm-thick m-dPhen:Ag (10 wt%) led to WF reduction for ITO from 4.6 to 3.1 eV and Ag from 4.5 to 3.5 eV, respectively, as illustrated in Fig. 4f. Since thermal evaporation enables the formation of smooth and homogeneous thin layers at the nanoscale[37], the WF downshifts of Al cathodes modified by Ag-doped EILs were nearly saturated when the thickness exceeded 5 nm[23,38] (Supplementary Fig. 13). Therefore, it was inferred that m-dPhen:Ag (5 nm, 10 wt%) can act as an efficient and universal EIL for optoelectronic devices with different electrodes. We also tested the ability of m-dPhen:Ag as an EIL for different ETMs. It is found that m-dPhen:Ag works well for conventional 1,3,5-tri(phenyl-2-benzimidazoly)-benzene (TPBi) with a high LUMO of $-2.7$ eV and possesses an $\eta_{\text{injection}}$ value of ~20% (Supplementary Fig. 14a)[39]. Moreover, for nitrogen-free polycyclic aromatic hydrocarbon (PAH)-type ETMs such as 9-($\alpha$-naphthyl)-10-($\beta$-naphthyl)-anthracene ($\alpha,\beta$-ADN)[40], electron injection is rather intractable for commonly used LiF and Liq, of which widespread application is limited due to coordination mechanisms relying on both Al cathodes and specific ETMs with nitrogen-containing heterocycles[18,41]. Notably, the EILs of m-dPhen:Ag allows efficient electron injection even for PAH-type ETMs (Supplementary Fig. 14b), which is different from the typical EILs of lithium compounds. The independence of CAN on cathodes and ETMs is superior to that of the widely used LiF, verifying the validity and compatibility of CAN strategies.

**Fabrication and performance of OLEDs**. These vacuum-deposited CAN-type EILs possessed high transmittance above 95% over the whole visible region, suggesting that the Ag-doped diphenanthroline ligands could function as highly transparent charge injection layers (Supplementary Fig. 16). Consequently, the performance of these Ag-doped EILs (5 nm) was systematically evaluated in OLEDs. Reference devices based on EILs of 1-nm LiF and 1-nm Liq were also fabricated for comparison. Devices with the structure of ITO/HATCN (5 nm)/NPB (30 nm)/BCzPh (10 nm)/$\alpha,\beta$-ADN:t-DABNA (30 nm, 2 wt%)/CzPhPy (10 nm)/DPPyA (20 nm)/EIL (1 or 5 nm)/Al (150 nm) were fabricated, and the energy level alignment around the cathode is plotted in Fig. 5b. Herein, HATCN, NPB, BCzPh and CzPhPy are 1,4,5,8,9,11-hex-aazatriphenylene-hexacarbonitrile, 4,4′-N,N′-bis[N-(1-naphthyl)-N-phenylamino]biphenyl, N,N′-diphenylbicarbazole and 4,6-bis(3-(9H-carbazol-9-yl)phenyl)pyrimidine, respectively. 2,12-Di-tert-butyl-5,9-bis(4-(tert-butyl)phenyl)-5,9-dihydro-5,9-diaza-13b-bor-anaphtho[3,2,1-de]anthracene (t-DABNA) was employed as a deep-blue fluorescent emitter (Supplementary Fig. 17). All devices exhibited identical electroluminescence spectra at 461 nm with a narrow full-width at half-maximum (FWHM) of 25 nm and CIE coordinates of (0.133, 0.085), confirming that the pure-blue emission originated from the t-DABNA dopants. The results of the $J-V$ characteristics in OLEDs agreed well with their electron injection efficiencies in EODs, and the highest current densities were obtained from the device based on m-dPhen:Ag. This device exhibited the lowest driving voltage of 4.0 V at a current density of 10 mA cm$^{-2}$, which is approximately 0.2 V lower than those for EILs of LiF, Liq and m-dPPhen:Ag. Correspondingly, the best device with m-dPhen:Ag exhibited the lowest driving voltages of 4.2, 5.4, and 6.0 V at 1000, 5000, and 10,000 cd m$^{-2}$, respectively.

Notably, the device with m-dPhen:Ag as the EIL displayed the highest maximum EQE (EQE$_{\text{max}}$) of 10.3% and low-efficiency roll-off with high EQEs of 10.3% at 1000 cd m$^{-2}$ (EQE$_{1000}$), 9.5% at 5000 cd m$^{-2}$ (EQE$_{5000}$), and 9.0% at 10,000 cd m$^{-2}$ (EQE$_{10000}$). In contrast, other devices showed much inferior efficiencies, with EQE$_{\text{max}}$ values of 7.8, 7.8, and 6.9% for m-dPPhen:Ag, LiF, and

Liq, respectively (Fig. 5e). To the best of our knowledge, the EQE$_{\text{max}}$ and EQE$_{1000}$ of the device with m-dPhen:Ag, which remained over 10%, are among the highest values ever reported for deep-blue OLEDs based on TTA (Supplementary Table 2). Furthermore, the hole/electron currents of single-carrier devices were characterized to study the charge balance of the devices based on different EILs. As shown in Supplementary Fig. 19, the LiF-based devices suffered from insufficient electron injection, and the electrons were the minority carriers in the emitting layers thereof. It has been demonstrated that enhanced charge carrier injection and improved carrier balance in the emitting layers could promote charge recombination characteristics as well as triplet harvesting by the TTA process[42]. Therefore, incorporating a more efficient EIL of m-dPhen:Ag would be beneficial to the device's performance. Owing to both high EQE and low driving voltages, the m-dPhen-based device achieved a high maximum power efficiency of 6.6 lm W$^{-1}$. Additionally, the power efficiency remains 4.4 lm W$^{-1}$ at 5000 cd m$^{-2}$ and 3.7 lm W$^{-1}$ at 10,000 cd m$^{-2}$. By replacing the emitting layer with phosphorescent dopants and thermally activated delayed fluorescent (TADF) materials, high EQEs exceeding 30% and improved power efficiency were attained for blue, green, and red OLEDs due to the enhanced electron injection, which suggested that incorporating efficient electron injection layers based on CAN also contribute to high-performance OLEDs (Supplementary Fig. 20 and Supplementary Table 3).

The stabilities of those TTA devices with various EILs were also evaluated at an initial luminance of 2000 cd m$^{-2}$. LT90 values (time to 90% of the initial luminance) of 79.2, 74.5, 26.0, and 18.5 h were obtained for the devices based on m-dPhen:Ag, m-dPPhen:Ag, LiF, and Liq as EILs, respectively. The longest lifetime of the device with m-dPhen:Ag can be interpreted as an improved carrier balance on account of the best electron injection (Fig. 5f). Interestingly, with similar charge injection, the device with m-dPPhen:Ag also showed an almost three times longer lifetime than the device with LiF. Plausible reasons for this phenomenon include the different moisture stabilities of EILs[4]. We measured the water contact angles of these EILs to investigate their moisture resistance (Supplementary Fig. 25). The Al cathodes modified by 5 nm Ag-doped diphenanthroline derivatives (10 wt%) showed obvious hydrophobic characteristics with high water contact angles in the range of 70°−80°. In contrast, the Al cathode with 1 nm LiF showed a low contact angle of 17.4°, which suggests that the high moisture affinity of LiF may lead to dark spots and degradation of OLEDs. In addition, previous works have revealed that metallic Li easily diffuses through organic layers, which leads to severe exciton quenching in the emitting layers and results in device degradation[10,43]. For EILs with CAN, the coordination effect of the Phen moieties will prevent the diffusion of Ag, thus improving the device lifetime[44].

In addition to bottom-emitting devices, we also studied CAN in a commercialized top-emitting OLED structure[45,46] with a reference device utilizing an Mg:Ag (1:9) cathode and Yb (1 nm) as EIL[47–49]. To simplify the fabrication process, a thin layer of pristine m-dPhen (1 nm) was used as the EIL hereby in situ coordination with Ag at the EIL/cathode interface. The electroluminescence performance of these top-emitting devices is shown in Fig. 6. The device with pristine m-dPhen (1 nm) as the EIL exhibited almost the same $J-V$ $-L$ characteristics as those of the Yb-based device (Fig. 6a), suggesting its electron injection ability. Since the current efficiency of deep-blue OLEDs with a narrow FWHM strongly depends on the color coordinate, the CCE value, which was calculated by dividing the current efficiency (cd A$^{-1}$) by the CIEy color coordinate, was adopted to evaluate the efficiency of deep-blue devices in the previous reports[48–52]. Figure 6b reveals similar CCE values for both devices, which are in agreement with their almost identical $J-V-L$

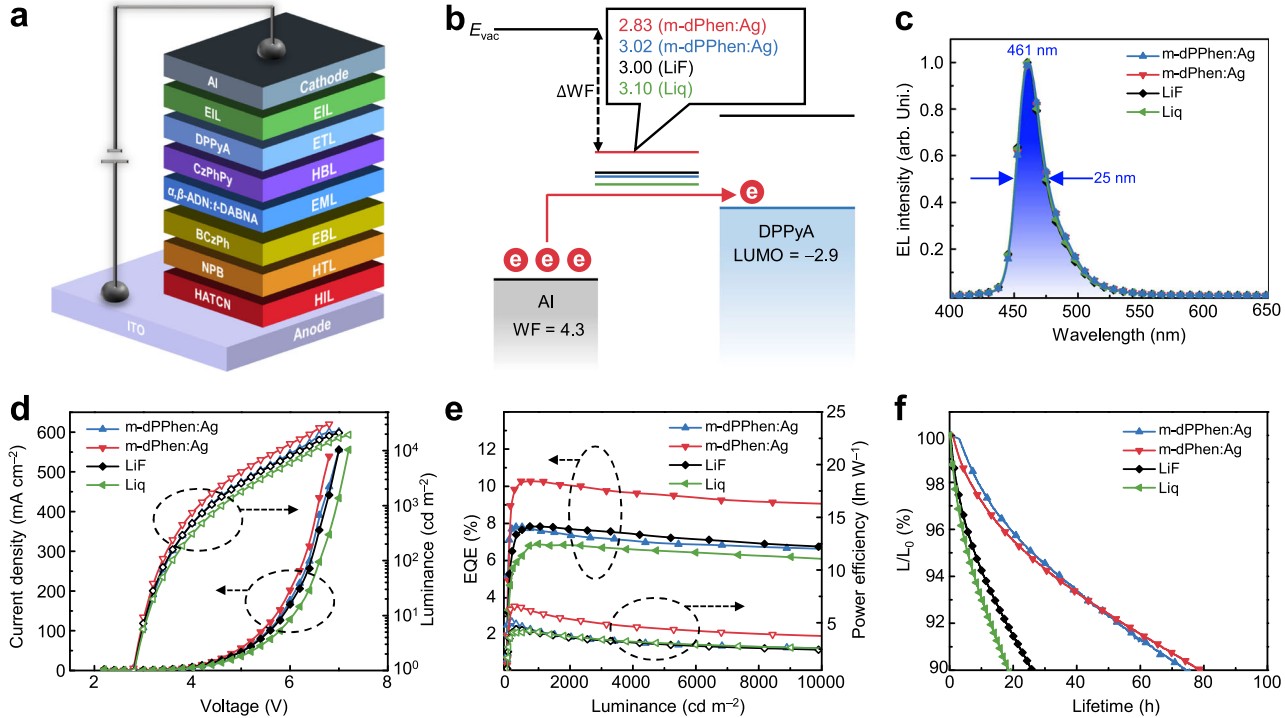

**Fig. 5 Summary of deep-blue OLEDs based on different EILs. a** The device structure of deep-blue OLEDs. **b** Energy level alignment of the EILs of LiF (1 nm), Liq (1 nm), m-dPPhen:Ag (5 nm, 10 wt%), and m-dPhen (5 nm, 10 wt%)/DPPyA in devices. **c** The EL spectra under 1000 cd m$^{-2}$. **d** The current density −voltage−luminance ($J$−$V$−$L$) of the OLEDs using different EILs of LiF (1 nm), Liq (1 nm), m-dPPhen:Ag (5 nm, 10 wt%), and m-dPhen (5 nm, 10 wt%) for comparison. **e** EQE−power efficiency−luminance of the OLEDs using different EILs. **f** The lifetimes measured at an initial luminance of 2000 cd m$^{-2}$.

characteristics. At a luminance of 1000 cd m$^{-2}$, the m-dPhen-based device displayed a low driving voltage of 4.03 V, CIE coordinates of (0.139 and 0.045), and a high CCE of 237 cd A$^{-1}$. The high CCE at a practical luminance of 1000 cd m$^{-2}$ is not only higher than that of typical TTA-based devices but also comparable to the performance of state-of-the-art deep-blue OLEDs (CIEy < 0.2) based on thermally activated delayed fluorescence-sensitized-fluorescence mechanisms (Supplementary Tables 4, 5). Remarkably, at a high initial luminance of 5000 cd m$^{-2}$, an LT95 of 104.1 h was obtained for the device with m-dPhen, which is even longer than that of the device with Yb (94.1 h), possibly owing to the eliminated metal diffusion with the m-dPhen EIL.

## Discussion

In conclusion, to eliminate the mutual exclusion between the high nucleophilic quality and high EA of mono-Phen ligands that limits the electron injection efficiency of CAN, we proposed meta-linked diphenanthroline-type derivatives as efficient ET ligands. These ligands not only possess a large EA with good electron transport ability but also favors the formation of tetrahedrally coordinated double-helical metal complexes to lower the ionization energy of air-stable metals. By optimizing the ligand structure, EILs with wide ranges of thickness and doping concentrations were obtained, which were also applicable for a variety of ETMs, electrodes, and device structures with injection efficiencies outperforming those of LiF and Liq. The top-emitting deep-blue OLEDs with an m-dPhen ligand as an EIL even achieved comparable device efficiency and better stability than those with a commercialized EIL of Yb. A CCE of ~240 cd A$^{-1}$ together with a superlong LT95 of over 100 h at 5000 cd m$^{-2}$ were recorded, satisfying the requirements of commercialization. We believe that further development of high-performance ligands based on CAN strategies may pave the way towards replacing reactive metals in the OLED industry.

## Methods

**Quantum chemical calculations**. The geometries of materials in the ground state and electrostatic potential maps were calculated with Gaussian 09 program package. The theoretical calculations were performed by using density functional theory (DFT) at the B3LYP/6-31 G(d) level.

**Fabrication and characterization of devices**. All devices were fabricated on ITO substrates with sheet resistances in the range of 20−80 Ω/square. Before the fabrication of devices, the ITO substrates were cleaned and followed by UV-ozone treatments. The organic layers are depositing onto the ITO substrate with rates varying from 0.5−1.0 Å s$^{-1}$ under the pressure of 5 × 10$^{-4}$ Pa. The deposition rate of the aluminum cathode is 2.0 Å s$^{-1}$. Quartz crystal sensors are employed to monitor the deposition rates and film thicknesses in situ. To fabricate CAN-type EIL by vacuum deposition, the deposition rate of silver is as slow as 0.01 Å s$^{-1}$ to secure efficient in situ coordination, and the diphenanthroline ligands are co-evaporated at rates of 2.0, 1.0, and 0.5 Å s$^{-1}$, thus achieving specific doping concentrations of 5, 10, 20 wt%. After vacuum deposition of functional layers, devices were transferred to a glovebox filled with nitrogen and then encapsulated with UV adhesive and glass covers. The device structure of bottom-emitting OLEDs is ITO/HATCN (5 nm)/NPB (30 nm)/BCzPh (10 nm)/α,β-ADN:t-DABNA (30 nm, 2 wt%)/CzPhPy (10 nm)/ETM (20 nm)/EIL (5 nm)/Al (150 nm). The device structure of top-emitting OLEDs is Ag/HIL (HT:p-dopant)/HTL/EBL/BH:BD/HBL/ETM:Liq/EIL (Yb or m-dPhen, 1 nm)/Mg:Ag (1:9)/CPL. The current density–luminance–voltage characteristics of bottom-emitting OLEDs and EODs were measured with the integration of a Keithley 2400 source meter and C9920-12 absolute EQE measurement system from Hamamatsu Photonics K.K., Japan.

**Ultraviolet photoelectron spectroscopy (UPS) measurement**. The UPS measurement was carried out by an Axis Ultra DLD (Kratos, UK) spectrometer with a HeI (21.22 eV) excitation source and a pass energy of 5 eV. A bias voltage of −9 V was applied to the sample during measurement for obtaining the cut-off region of secondary electrons. The Fermi level was identified by using Au as a reference.

**Reporting Summary**. Further information on research design is available in the Nature Research Reporting Summary linked to this article.

## Data availability

The data that support the findings of this study are available from the corresponding author upon reasonable request. The X-ray crystallographic data for the single-crystal structures reported in this study have been submitted to the Cambridge Crystallographic

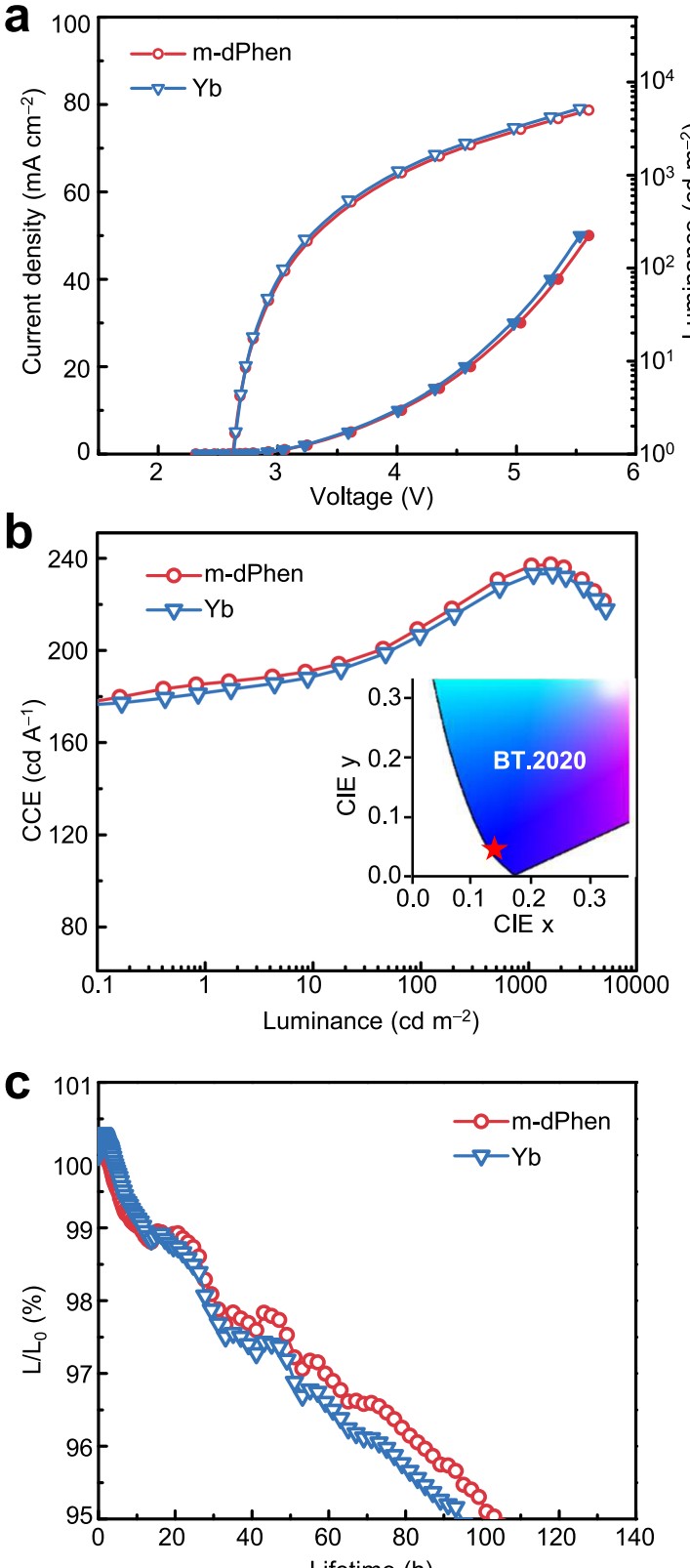

**Fig. 6 Performance of top-emitting OLEDs with EILs of Yb (1 nm) and m-dPhen (1 nm). a** Current density-voltage luminance ($J-V-L$) of the OLEDs using Yb and m-dPhen as EILs for comparison. **b** The CCE−luminance curves of the OLEDs using different EILs. **c** The lifetimes measured at an initial luminance of 5000 cd m$^{-2}$.

Data Centre (CCDC). The crystallographic data generated in this study are provided in Supplementary Data 1.

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

## Acknowledgements

This work was supported by the National Key Basic Research and Development Program of China (Grant Nos. 2017YFA0204501 and 2020YFA0715000), the National Natural Science Foundation of China (Grant Nos. 51903137 and 61890942), and Foshan Xianhu Laboratory of the Advanced Energy Science and Technology Guangdong Laboratory XHT2020-005. D.D. Zhang also thanks the financial support of the Young Elite Scientist Sponsorship Program (2019QNRC001) by the China Association for Science and Technology. The authors also wish to thank Prof. Chong Li of Nanjing Tech University for his support and help in the fabrication of the top-emitting OLEDs.

## Author contributions

L.D. and D.Z. jointly conceived and supervised the project. Z.L. conducted most part of the experiments. X.L. and Y.L. helped in theoretical calculation and ligand synthesis. C.Z.

and Y.Z. measured the single-crystal structures. T.H. carried out NMR measurement. All authors analyzed the results and wrote the manuscript.

## Competing interests

The authors declare no competing interests.
