## [Peer Review File · Nature Communications]

REVIEWER COMMENTS first round -

Reviewer #1 (Remarks to the Author):

Efficient n-doping techniques are essential for organic optoelectronics. Though numerous researches in this field, an effective n-doping strategy to replace highly-reactive metals remain an exigent task, particularly for the OLED industry. The same group previously reported an advanced concept of in situ coordination-activated n-doping (CAN) technique by utilizing air-stable metals as n-dopants of organic ligands with coordination sites. This CAN technique is vacuum-deposited, byproduct-free and widely applicable, possessing great potential to the revolution of n-doping technique in OLEDs. For CAN, the organic ligand is crucial for electron injection efficiency. Aiming at solving the trade-off between nucleophilic quality and electron affinity of the present derivatives, in this successive work, the authors creatively proposed meta-linkage diphenanthroline type ligands (m-dPhen) for CAN. The novel electron injection layer (EIL) based on m-dPhen:Ag enables formation of tetrahedrally coordinated double-helical metal complexes and efficient electron injection for various cathodes and electron transport materials, which is critical to the widespread application. Additionally, the corresponding deep-blue fluorescent OLEDs achieved impressive CCE of 237 cd A⁻¹ at 1000 cd m⁻² and record-high T95 > 100 h at 5000 cd m⁻², satisfying the requirement for industrial application.

This work deepens the comprehensive understanding of CAN strategies and the ligands proposed here may bring a revolutionary change to the OLED industry as well as other organic optoelectronics. And also this work is well organized and written. Based on the above discussions, I strongly recommend the acceptance of this manuscript after some revisions below:

- (1) The full names of compounds such as BCzPh and CzPhPy in deep-blue devices should be provided when used for the first time and their energy levels should also be plotted.
- (2) As seen in Fig. 4, the device with EIL of m-dPhen:Ag showed highest EQE among all devices. Given that the balance of charge injection/transport has great impact on the device performances, the authors should provide a brief explanation for the superior device performances with m-dPhen.
- (3) The performances of OLEDs based on non-doped m-dPhen as EIL should be added to verify the effect of CAN on electron injection.
- (4) The doping process should be described in detail to improve data reproducibility.
- (5) The optical transmittance of these Ag-doped EILs should be provided, which is one of the key factors for charge injection layers.
- (6) The UPS data and shift of HOMO position is important to confirm the n-doping effect. The authors should mark the energy level shifts of pristine ligands and Ag-doped ligands on the Supplementary Fig. 13.
- (7) Did the author verify if CAN techniques can be applied to improve electron extraction for other photovoltaic devices?
- (8) The Figs. 1c, 1d, 4a and 5b are not clear. Figure legends should be added in Fig. 4c.

Reviewer #2 (Remarks to the Author):

This manuscript by Duan et al. demonstrated that efficient and stable electron injection can be achieved by incorporating novel diphenanthroline type ligands for in situ coordination-activated n-doping (CAN) techniques. The designed meta-linkage molecule (m-dPhen) shows high electron affinity with deep LUMO level (~3.0 eV) and strong nucleophilic quality by forming tetrahedrally coordinated double-helical structure, thus enabling high electron injection efficiency. The material design strategy of meta-linked diphenanthroline ligands proposed here successfully circumvents the limitations of the mono-Phen type ligands in previous reports of the same group as well as others, which are quite impressive and rigorous. And the results and discussions will surely impact future studies on this advanced n-doping technology. More importantly, the resulting deep-blue OLEDs shows high CCE of 237 cd/A and super-long lifetime with T95 of 104.1 h at 5000 cd/m², which are obviously the one of best results ever reported for deep-blue OLEDs. Overall, the manuscript presented an impressive advance for n-doping technique, which should also interest

other organic electronic devices besides OLEDs. This manuscript is well written and topics are meaningful. Therefore, I think the paper could be accepted after further improvement.

(1) In the manuscript, the meta-linkage diphenanthroline type ligands are superior to mono-type Phen ligands for CAN strategies, an in-depth comparison and explanation should be added to present the advantages of proposed designing strategies.

(2) The authors confirmed the validity and compatibility of CAN in different device structures in supporting information. Apart from current density-voltage-brightness characteristics, the EL spectra of each OLEDs should be given to confirm the cavity effect.

(3) The items and figure legends in figures 1, 4 and 5 should be modified to give clear information.

(4) The device performances of deep-blue OLEDs based on three diphenanthroline ligands should be summarized in Table.

(5) The authors should give a brief explanation why the performances of deep-blue OLED with m-dPhen and Yb are compared by CCE.

Reviewer #3 (Remarks to the Author):

This manuscript employed the CAN strategy depending on metal-linked diphenanthroline-type ligands, co-evaporated with Silver, as n-type dopants to enhance electron injection efficiencies and thus achievement of high-efficiency deep-blue OLED with a high current efficiency (calibrated by y-color coordinate (0.0045) of 237 cdA⁻¹ and superb LT95 of 104.1 hours at 5000 cdm⁻²). However, to improve the current manuscript, I believe the following comments should be addressed:

1) The authors claim that the high electron injection efficiency is independent on the n-type dopant concentration of Ag. In Supplementary Figure 11(a), the authors compare the varied concentrations of 5, 10, 20 wt% to 0 wt% and the results displayed significantly enhanced current densities over the 0 wt% device. However, low dopant concentrations like 1 wt% or 3 wt% should be added to further investigate electron injection efficiencies with no impact placed by such low dopant concentrations when considering n-type doping as electrical doping. In addition, the data for the 0 wt% device should be added to Supplementary Figure 11(b).

2) The authors built one set of EODs employing different EILs, including Ag-doped diphenanthroline ligands, commonly-used LiF and Liq with the same device structure (Figure 3c). However, the thickness of the EILs varied from 5 nm to 1 nm before investigation of electron injection efficiencies independent on the EIL thickness. Logically, the same thickness of EILs should be maintained (1 nm) for a solid "apples to apples" comparison while focusing on one variable at a time. Despite the achievement of an excellent result about the high electron injection efficiency independent on thickness of EILs in Figure 3e, the design of the EIL thickness has nonetheless remained different regardless of the consideration of the typical thickness range from 0.5 nm to 2 nm for EILs such as LiF and Liq.

3) The UPS characterizations prove that the work functions of different cathode electrodes have been reduced due to the deposition of a 5-nm-thick m-dphen:Ag (10 wt%) n-dopant, which has modified the interface of cathodes. Forrest et al. had reported that the smooth and homogeneous surface of nano-scale films can be achieved via thermal evaporation, a typical deposition technique for OLEDs fabrication (Chem. Rev. 1997, 97, 1793-1896). Moreover, to our knowledge, sensitivity of UPS source is limited within a few nanometers less than 5 nm. Theoretically, only the Fermi level can be characterized for the shallow surface of 5 nm-thick films rather than the work function of the bulk cathode material. However, the results definitely demonstrate the reduced work function of individual cathodes upon top-deposition of a 5-nm-thick m-dphen:Ag on the cathode electrodes. Nevertheless, I would like the authors to please provide the result of calibrated thickness of the co-evaporated EILs and measure their intrinsic Fermi levels by depositing thick (> 20 nm) films using UPS to further support their conclusions.

4) The calculated IEs for Ag(m-dphen) and Ag((m-dphen)₂) denoted as 3.45 eV and 3.06 eV in the main manuscript text, which are different from the values displayed in Figure 3b.

5) Minor: A few syntax errors and typos are found by this reviewer. Examples:

- Abstract line 9: is achieved;
- Results line 97: Different from;
- Results line 145: (UPS) are.

To be accepted by Nature Communications, the reviewer believes that any similar errors or typos should be addressed.

Reviewer #4 (Remarks to the Author):

This manuscript concerns itself with coordination-activated doping using organic ligands. The authors describe a fundamental problem with this approach, namely that the ligand must be nucleophilic to form the bond, but should also have a high electron affinity for efficient n-doping. A chelating ligand is shown to overcome this issue, and highly efficient OLEDs are manufactured using this electron transport layer.

The synthesized materials, layers and devices are carefully characterized by a very large number of techniques (maybe even a confusingly large number of techniques, but if it fits within length restrictions of the manuscript that is OK). The methodology appears sound.

The motivation for this work is in a large part taken from a desire to compete with current industry standards for OLED efficiency. For example, it is stated that "The OLEDs industry with multi-billion outputs today still relies on highly reactive metals to ensure desirable device performances despite of hazardous reactivity and diffusivity." No citation is attached to this statement, and moreover, it is hard to know that because of likely trade-secrets. The potential impact of this work is therefore difficult to judge for me. There is an informative table (Table 2) in the supplementary materials. From that table it appears that the progress made here is not particularly significant in terms of efficiency. I would like the authors to comment on this point.

Regarding the characterization of the OLED devices, the authors state "The performances of EILs consisted of 10 wt% Ag-doped ligands (5 nm) were systematically evaluated in OLEDs with 1-nm LiF and 1-nm Liq as references." I hope this means that LiF reference devices were compared to devices made from the materials proposed here, and that the latter devices did not contain LiF. Is this correct? The authors should state this more clearly. Otherwise LiF in one form would be compared to LiF in another form, and not to the pure materials proposed here.

Another question I have is why were TTA devices studied, wouldn't it have been more straightforward (and scientifically clearer) to study charge injection efficiency in direct charge recombination emitters, rather than the indirect TTA process?

After receiving the authors' responses I will make a recommendation regarding publication.

Reviewers' Comments

Reviewer #1 (Remarks to the Author):

Efficient n-doping techniques are essential for organic optoelectronics. Though numerous researches in this field, an effective n-doping strategy to replace highly-reactive metals remain an exigent task, particularly for the OLED industry. The same group previously reported an advanced concept of in situ coordination-activated n-doping (CAN) technique by utilizing air-stable metals as n-dopants of organic ligands with coordination sites. This CAN technique is vacuum-deposited, byproduct-free and widely applicable, possessing great potential to the revolution of n-doping technique in OLEDs. For CAN, the organic ligand is crucial for electron injection efficiency. Aiming at solving the trade-off between nucleophilic quality and electron affinity of the present derivatives, in this successive work, the authors creatively proposed meta-linkage diphenanthroline type ligands (m-dPhen) for CAN. The novel electron injection layer (EIL) based on m-dPhen:Ag enables formation of tetrahedrally coordinated double-helical metal complexes and efficient electron injection for various cathodes and electron transport materials, which is critical to the widespread application. Additionally, the corresponding deep-blue fluorescent OLEDs achieved impressive CCE of 237 cd A^{-1} at 1000 cd m^{-2} and record-high T95 $> 100 \text{ h}$ at 5000 cd m^{-2} , satisfying the requirement for industrial application.

This work deepens the comprehensive understanding of CAN strategies and the ligands proposed here may bring a revolutionary change to the OLED industry as well as other organic optoelectronics. And also this work is well organized and written. Based on the above discussions, I strongly recommend the acceptance of this manuscript after some revisions below:

- (1) The full names of compounds such as BCzPh and CzPhPy in deep-blue devices should be provided when used for the first time and their energy levels should also be plotted.
- (2) As seen in Fig. 4, the device with EIL of m-dPhen:Ag showed highest EQE among all devices. Given that the balance of charge injection/transport has great impact on the device performances, the authors should provide a brief explanation for the superior device performances with m-dPhen.
- (3) The performances of OLEDs based on non-doped m-dPhen as EIL should be added to verify the effect of CAN on electron injection.
- (4) The doping process should be described in detail to improve data reproducibility.
- (5) The optical transmittance of these Ag-doped EILs should be provided, which is one of the key factors for charge injection layers.
- (6) The UPS data and shift of HOMO position is important to confirm the n-doping effect. The authors should mark the energy level shifts of pristine ligands and Ag-doped ligands on the Supplementary Fig. 13.
- (7) Did the author verify if CAN techniques can be applied to improve electron extraction for other photovoltaic devices?
- (8) The Figs. 1c, 1d, 4a and 5b are not clear. Figure legends should be added in Fig. 4c.

Reviewer #2 (Remarks to the Author):

This manuscript by Duan et al. demonstrated that efficient and stable electron injection can be achieved by incorporating novel diphenanthroline type ligands for in situ coordination-activated n-doping (CAN) techniques. The designed meta-linkage molecule (m-dPhen) shows high electron affinity with deep LUMO level (~ 3.0 eV) and strong nucleophilic quality by forming tetrahedrally coordinated double-helical structure, thus enabling high electron injection efficiency. The material design strategy of meta-linked diphenanthroline ligands proposed here successfully circumvents the limitations of the mono-Phen type ligands in previous reports of the same group as well as others, which are quite impressive and rigorous. And the results and discussions will surely impact future studies on this advanced n-doping technology. More importantly, the resulting deep-blue OLEDs shows high CCE of 237 cd/A and super-long lifetime with T95 of 104.1 h at 5000 cd/m², which are obviously the one of best results ever reported for deep-blue OLEDs. Overall, the manuscript presented an impressive advance for n-doping technique, which should also interest other organic electronic devices besides OLEDs. This manuscript is well written and topics are meaningful. Therefore, I think the paper could be accepted after further improvement.

(1) In the manuscript, the meta-linkage diphenanthroline type ligands are superior to mono-type Phen ligands for CAN strategies, an in-depth comparison and explanation should be added to present the advantages of proposed designing strategies.

(2) The authors confirmed the validity and compatibility of CAN in different device structures in supporting information. Apart from current density-voltage-brightness characteristics, the EL spectra of each OLEDs should be given to confirm the cavity effect.

(3) The items and figure legends in figures 1, 4 and 5 should be modified to give clear information.

(4) The device performances of deep-blue OLEDs based on three diphenanthroline ligands should be summarized in Table.

(5) The authors should give a brief explanation why the performances of deep-blue OLED with m-dPhen and Yb are compared by CCE.

Reviewer #3 (Remarks to the Author):

This manuscript employed the CAN strategy depending on metal-linked diphenanthroline-type ligands, co-evaporated with Silver, as n-type dopants to enhance electron injection efficiencies and thus achievement of high-efficiency deep-blue OLED with a high current efficiency (calibrated by y-color coordinate (0.0045) of 237 cdA⁻¹ and superb LT95 of 104.1 hours at 5000 cdm⁻²). However, to improve the current manuscript, I believe the following comments should be addressed:

1) The authors claim that the high electron injection efficiency is independent on the n-type dopant concentration of Ag. In Supplementary Figure 11(a), the authors compare

the varied concentrations of 5, 10, 20 wt% to 0 wt% and the results displayed significantly enhanced current densities over the 0 wt% device. However, low dopant concentrations like 1 wt% or 3 wt% should be added to further investigate electron injection efficiencies with no impact placed by such low dopant concentrations when considering n-type doping as electrical doping. In addition, the data for the 0 wt% device should be added to Supplementary Figure 11(b).

2) The authors built one set of EODs employing different EILs, including Ag-doped diphenanthroline ligands, commonly-used LiF and Liq with the same device structure (Figure 3c). However, the thickness of the EILs varied from 5 nm to 1 nm before investigation of electron injection efficiencies independent on the EIL thickness. Logically, the same thickness of EILs should be maintained (1 nm) for a solid “apples to apples” comparison while focusing on one variable at a time. Despite the achievement of an excellent result about the high electron injection efficiency independent on thickness of EILs in Figure 3e, the design of the EIL thickness has nonetheless remained different regardless of the consideration of the typical thickness range from 0.5 nm to 2 nm for EILs such as LiF and Liq.

3) The UPS characterizations prove that the work functions of different cathode electrodes have been reduced due to the deposition of a 5-nm-thick m-dphen:Ag (10 wt%) n-dopant, which has modified the interface of cathodes. Forrest et al. had reported that the smooth and homogeneous surface of nano-scale films can be achieved via thermal evaporation, a typical deposition technique for OLEDs fabrication (Chem. Rev. 1997, 97, 1793-1896). Moreover, to our knowledge, sensitivity of UPS source is limited within a few nanometers less than 5 nm. Theoretically, only the Fermi level can be characterized for the shallow surface of 5 nm-thick films rather than the work function of the bulk cathode material. However, the results definitely demonstrate the reduced work function of individual cathodes upon top-deposition of a 5-nm-thick m-dphen:Ag on the cathode electrodes. Nevertheless, I would like the authors to please provide the result of calibrated thickness of the co-evaporated EILs and measure their intrinsic Fermi levels by depositing thick (> 20 nm) films using UPS to further support their conclusions.

4) The calculated IEs for Ag(m-dphen) and Ag((m-dphen)₂) denoted as 3.45 eV and 3.06 eV in the main manuscript text, which are different from the values displayed in Figure 3b.

5) Minor: A few syntax errors and typos are found by this reviewer. Examples:

- Abstract line 9: is achieved;
- Results line 97: Different from;
- Results line 145: (UPS) are.

To be accepted by Nature Communications, the reviewer believes that any similar errors or typos should be addressed.

Reviewer #4 (Remarks to the Author):

This manuscript concerns itself with coordination-activated doping using organic ligands. The authors describe a fundamental problem with this approach, namely that the ligand must be nucleophilic to form the bond, but should also have a high electron affinity for efficient n-doping. A chelating ligand is shown to overcome this issue, and highly efficient OLEDs are manufactured using this electron transport layer.

The synthesized materials, layers and devices are carefully characterized by a very large number of techniques (maybe even a confusingly large number of techniques, but if it fits within length restrictions of the manuscript that is OK). The methodology appears sound.

The motivation for this work is in a large part taken from a desire to compete with current industry standards for OLED efficiency. For example, it is stated that “The OLEDs industry with multi-billion outputs today still relies on highly reactive metals to ensure desirable device performances despite of hazardous reactivity and diffusivity.” No citation is attached to this statement, and moreover, it is hard to know that because of likely trade-secrets. The potential impact of this work is therefore difficult to judge for me. There is an informative table (Table 2) in the supplementary materials. From that table it appears that the progress made here is not particularly significant in terms of efficiency. I would like the authors to comment on this point.

Regarding the characterization of the OLED devices, the authors state “The performances of EILs consisted of 10 wt% Ag-doped ligands (5 nm) were systematically evaluated in OLEDs with 1-nm LiF and 1-nm Liq as references.” I hope this means that LiF reference devices were compared to devices made from the materials proposed here, and that the latter devices did not contain LiF. Is this correct? The authors should state this more clearly. Otherwise LiF in one form would be compared to LiF in another form, and not to the pure materials proposed here.

Another question I have is why were TTA devices studied, wouldn't it have been more straightforward (and scientifically clearer) to study charge injection efficiency in direct charge recombination emitters, rather than the indirect TTA process?

After receiving the authors' responses I will make a recommendation regarding publication.

Responses to the Reviewers

Reviewer #1

General comment: Efficient n-doping techniques are essential for organic optoelectronics. Though numerous researches in this field, an effective n-doping strategy to replace highly-reactive metals remain an exigent task, particularly for the OLED industry. The same group previously reported an advanced concept of in situ coordination-activated n-doping (CAN) technique by utilizing air-stable metals as n-dopants of organic ligands with coordination sites. This CAN technique is vacuum-deposited, byproduct-free and widely applicable, possessing great potential to the revolution of n-doping technique in OLEDs. For CAN, the organic ligand is crucial for electron injection efficiency. Aiming at solving the trade-off between nucleophilic quality and electron affinity of the present derivatives, in this successive work, the authors creatively proposed meta-linkage diphenanthroline type ligands (m-dPhen) for CAN. The novel electron injection layer (EIL) based on m-dPhen:Ag enables formation of tetrahedrally coordinated double-helical metal complexes and efficient electron injection for various cathodes and electron transport materials, which is critical to the widespread application. Additionally, the corresponding deep-blue fluorescent OLEDs achieved impressive CCE of 237 cd A^{-1} at 1000 cd m^{-2} and record-high T95 > 100 h at 5000 cd m^{-2} , satisfying the requirement for industrial application.

This work deepens the comprehensive understanding of CAN strategies and the ligands proposed here may bring a revolutionary change to the OLED industry as well as other organic optoelectronics. And also this work is well organized and written. Based on the above discussions, I strongly recommend the acceptance of this manuscript after some revisions below:

Author reply: Thanks a lot for your valuable comments and suggestions. We have revised our manuscript according to your suggestions, as highlighted by red-characters in the revised manuscript.

Comment 1: The full names of compounds such as BCzPh and CzPhPy in deep-blue devices should be provided when used for the first time and their energy levels should also be plotted.

Response: Thank you for your suggestions. We have added the full names of all compounds and their energy levels were added as Supplementary Fig. 17 as follows:

“Herein, HATCN, NPB, BCzPh and CzPhPy are 1,4,5,8,9,11-hexaazatriphenylene-hexacarbonitrile, 4,4'-N,N'-bis[N-(1-naphthyl)-N-phenylamino]biphenyl, N,N'-diphenylbicarbazole and 4,6-bis(3-(9H-carbazol-9-yl)phenyl)pyrimidine, respectively. 2,12-ditert-butyl-5,9-bis(4-(tert-butyl)phenyl)-5,9-dihydro-5,9-diaza-13b-boranaphtho[3,2,1-de]anthracene (*t*-DABNA) was employed as the deep-blue fluorescent emitter.”

Supplementary Figure 17. (a) The chemical structures of materials used in bottom-emitting deep-blue OLEDs. (b) The energy levels of the deep-blue OLEDs

Comment 2: As seen in Fig. 4, the device with EIL of m-dPhen:Ag showed highest EQE among all devices. Given that the balance of charge injection/transport has great impact on the device performances, the authors should provide a brief explanation for the superior device performances with m-dPhen.

Response: Thank you for your constructive suggestions. Charge balance was critical to the carrier recombination and device performances for TTA-driven OLEDs. The superior performances of the devices with m-dPhen should be ascribed to the electron injection enhancement by using m-dPhen:Ag with the highest $\eta_{\text{injection}}$. Detailed analysis was summarized as follows:

(1) From the aspect of charge transport properties, NPB and DPPyA exhibited comparable carrier mobilities with μ_{h} of $2.7 \times 10^{-4} \text{ cm}^2 \text{ V}^{-1} \text{ s}^{-1}$ and μ_{e} of $4.6 \times 10^{-4} \text{ cm}^2 \text{ V}^{-1} \text{ s}^{-1}$

² V⁻¹ s⁻¹, respectively. For emissive layers, α,β -ADN with polycyclic aromatic hydrocarbon (PAH) skeleton have bipolar charge transport characters (*Isr. J. Chem.* 2012, 52, 484). However, the hole mobility ($\mu_h = 2.9 \times 10^{-5}$ cm² V⁻¹ s⁻¹) of BCzPh was about one order of magnitude higher than the electron mobility ($\mu_e = 3-5 \times 10^{-6}$ cm² V⁻¹ s⁻¹) of CzPhPy (*Chem. Mater.* 2011, 23, 274; *Org. Electron.* 2014, 15, 1368), which may lead to excessive holes in the emitting layers.

(2) From the aspect of charge injection, ITO/HATCN/NPB have been demonstrated to achieve an injection efficiency about 30% (*Adv. Funct. Mater.* 2012, 22, 3261), and common-used LiF (1 nm) showed a limited $\eta_{\text{injection}}$ of no more than 40% at 150 kV cm⁻¹.

Given the influence of charge injection/transport properties, it would be anticipated that holes were the majority carriers and electrons were the minority carriers for the deep-blue OLEDs with LiF. To further verify the charge balance experimentally, we measured the current density-electrical field characteristics of hole-only half-device (HOHD) and electron-only half-devices (EOHDs) as plotted in Supplementary Fig. 19:

Supplementary Figure 19. The current density-electrical field characteristics of hole-only half-devices (HOHDs) and electron-only half-devices (EOHDs). The device structure of HOHD (black) is ITO/HATCN (5 nm)/NPB (30 nm)/BCzPh (10 nm)/ α,β -ADN (30 nm)/Al (150 nm); The device structure of EOHDs are ITO/ α,β -ADN (30 nm)/CzPhPy (10 nm)/DPPyA (30 nm)/EIL (1 nm for LiF and Liq, 5 nm for Ag-doped p-dPPhen, m-dPPhen and m-dPhen)/Al (150 nm).

As illustrated in Supplementary Fig. 19, the current densities of HOHD were higher than those of LiF-based EOHDs at the same electrical fields, which suggested that electrons were the minority carriers for the reference deep-blue OLEDs due to insufficient electron injection of LiF/Al. While the current densities of EOHD with EIL of m-dPhen:Ag was close to the current densities of EOHD at the same electrical fields, which facilitates charge balance and superior device performances for the OLEDs with aforementioned structures. Based on these discussions, the analysis of the device performances was revised as follows:

“Furthermore, the hole/electron currents of single carrier devices were characterized to study the charge balance of the devices based on different EILs. As shown in Supplementary Fig. 19, the LiF-based devices suffered from insufficient electron injection and the electrons were the minority carriers in the emitting layers thereof. It has been demonstrated that enhanced charge carrier injection and improved carrier balance in the emitting layers could promote charge recombination characteristics as well as triplet harvesting by TTA process⁴². So incorporating a more efficient EIL of m-dPhen:Ag would be beneficial to the device performances.”

Supplementary Figure 19. The current density-electrical field characteristics of hole-only half-devices (HOHDs) and electron-only half-devices (EOHDs). The device structure of HOHD (black) is ITO/HATCN (5 nm)/NPB (30 nm)/BCzPh (10 nm)/ α,β -ADN (30 nm)/Al (150 nm); The device structure of EOHDs are ITO/ α,β -ADN (30 nm)/CzPhPy (10 nm)/DPPyA (30 nm)/EIL (1 nm for LiF and Liq, 5 nm for Ag-doped p-dPPhen, m-dPPhen and m-dPhen)/Al (150 nm).

Comment 3: The performances of OLEDs based on non-doped m-dPhen as EIL should be added to verify the effect of CAN on electron injection.

Response: Thank you for your nice suggestion, the performances of the OLED based on non-doped m-dPhen as EIL have been added as Supplementary Fig. 21. According to UPS measurement in Supplementary Fig. 15, the aluminum cathodes modified by non-doped diphenanthroline compounds possessed relatively high work function (WF) about 3.2 eV, thus leading to sizable injection barriers between the cathodes and electron transport materials. As a result, the OLEDs without silver as n-dopants exhibited high driving voltage and low external quantum efficiency as follows:

Supplementary Figure 21. (a) The current density-voltage-luminance (J - V - L) of OLED devices with EIL of non-doped m-dPhen (5nm). (b) The EQE-power efficiency-luminance of OLED devices with EIL of non-doped m-dPhen (5 nm).

Comment 4: The doping process should be described in detail to improve data reproducibility.

Response: Thanks very much for the comment and suggestion. The detailed description of doping process between Ag and diphenanthroline compounds at the doping concentrations of 5, 10, 20 wt% were revised as follows:

“Quartz crystal sensors are employed to monitor the deposition rates and film thicknesses in situ. To fabricate CAN-type EIL by vacuum deposition, the deposition rate of silver is as slow as 0.01 \AA s^{-1} to secure efficient in-situ coordination, and the diphenanthroline ligands are co-evaporated at rates of 2.0 \AA s^{-1} , 1.0 \AA s^{-1} , 0.5 \AA s^{-1} , thus achieving specific doping concentrations of 5, 10, 20 wt%.”

Comment 5: The optical transmittance of these Ag-doped EILs should be provided, which is one of the key factors for charge injection layers.

Response: Thank you for your suggestions. We measured the UV-vis absorption spectra and transmittances to study the of optical properties of these Ag-doped EILs. The Ag-doped films were deposited on quartz substrates with doping concentration of 10 wt% and thickness of 10 nm. As shown in Supplementary Fig. 16, these EILs consisting of Ag-doped different diphenanthroline ligands possessed high transmittances above 95% and flat curves in the wavelength range of 400–700 nm, which indicated the optical transmittance and chroma fidelity of the transmitted light for these CAN-type EILs could meet the requirement for organic light-emitting diodes. The following description of the optical properties of EILs was revised as follows:

“These vacuum-deposited CAN-type EILs possessed high transmittance above 95% in the whole visible region, suggesting that the Ag-doped diphenanthroline ligands could function as highly transparent charge injection layers. Consequently, the performances

of these Ag-doped EILs (5 nm) were systematically evaluated in OLEDs. And reference devices based on EILs of 1-nm LiF and 1-nm Liq were also fabricated for comparison.”

Supplementary Figure. 16 (a) The normalized absorption spectra of p-dPPhen:Ag (10 wt%, 10 nm), m-dPPhen:Ag (10 wt%, 10 nm) and m-dPhen:Ag (10 wt%, 10 nm), respectively. (b) The light transmittances of the p-dPPhen:Ag (10 wt%, 10 nm), m-dPPhen:Ag (10 wt%, 10 nm) and m-dPhen:Ag (10 wt%, 10 nm), respectively.

Comment 6: The UPS data and shift of HOMO position is important to confirm the n-doping effect. The authors should mark the energy level shifts of pristine ligands and Ag-doped ligands on the Supplementary Fig. 13.

Response: Thank you for your suggestions. The ultraviolet photoelectron spectroscopy (UPS) measurement was carried out to compare the energetic differences between HOMO level of organic layers and the Fermi level upon doping, thus confirming the n-doping effect. Here, the HOMO position is derived from the intersection of the two tangent lines in the low binding energy region according to previous report (*Nat. Mater.* 2012, 11, 76), which were marked in purple. To accurately determine the Fermi level shifts, the position of the HOMO was extrapolated based on the regions where the signals of organic layers increased remarkably. As was shown in Supplementary Fig. 15, upon doping with silver, the line shapes didn't change significantly, but the Fermi levels shifted away from the HOMO level, thus indicated the electrical n-doping effect (*Adv. Mater.* 2019, 31, 1904201). Accordingly, we marked the tangent in purple and modified Supplementary Fig. 15 as follows:

Supplementary Figure 15. Ultraviolet photoelectron spectroscopy (UPS) analysis and schematic energy-level diagrams of Al cathodes modified by 5 nm of pristine or Ag-doped diphenanthroline films. (a) UPS analysis and (b) schematic energy-level diagrams of Al (10 nm)/p-dPPhen (5 nm) and Al (10 nm)/p-dPPhen:Ag (5 nm, 10 wt%). (c) UPS analysis and (d) schematic energy-level diagrams of Al (10 nm)/m-dPPhen (5 nm) and Al (10 nm)/m-dPPhen:Ag (5 nm, 10 wt%). (e) UPS analysis and (f) Schematic energy-level diagrams of Al (10 nm)/m-dPhen (5 nm) and Al (10 nm)/m-dPhen:Ag (5 nm, 10 wt%). The HOMO positions were determined by the intersection of the two tangent lines (in purple) at the low binding energy region. According to the results of UPS analysis, the energy offsets between E_F and HOMO level increased after Ag-doping, which verifying the coordination-activated n-doping effect.

Comment 7: Did the author verify if CAN techniques can be applied to improve electron extraction for other photovoltaic devices?

Response: Thank you very much for your valuable comments and suggestions, which have greatly inspired us. Owing to the lack of fabrication and measurement systems for

other photovoltaic devices, it's challenging for us to verify the whether CAN techniques are applicable as electron extraction layers for other photovoltaic devices. Nevertheless, we believed that CAN technique is an available method to improve electron extraction for other photovoltaic devices based on the following considerations. Firstly, the LUMO levels of prevailing electron transport materials in organic solar cells (OSC) were mostly located in the range from -4 to -3 eV (*Adv. Energy Mater.* 2018, 8, 1800249). The CAN-modified electrodes such as Al and Ag showed low work function of 2.7 eV and 3.5 eV, which were favorable to efficient electron extraction from the perspective of energy alignment. Secondly, in the recent report, 2,9-dimethyl-4,7-diphenyl-1,10-phenanthroline (BCP):Ag complex were successfully employed to reduce the electron extraction barrier between a C₆₀ electron transport layer and indium-zinc oxide (IZO) top electrode in OSCs, thus enabling high device efficiency of 18.19% (*J. Mater. Chem. A* 2021, 9, 12009). Since the m-dPhen have been demonstrated to outperform typical mono-Phen ligands such as 4,7-dimethoxy-1,10-phenanthroline (p-MeO-Phen) by breaking the mutual exclusion between high nucleophilic quality and high electron affinity, it could be anticipated that CAN technique would also be applicable to cathode modification in OSCs. Moreover, the m-dPhen: Ag also exhibited good thermal stability and hydrophobic characters, which were expected to facilitate the operation stability of photovoltaic devices. The discussion about the potential of CAN technique for other photovoltaic devices were added as follows:

“By decoupling nucleophilic quality and electron affinity, the meta-linked diphenanthroline-type ligand, m-dPhen, outperformed typical mono-Phen ligands of p-MeO-Phen for electron injection. As a result of the strong chelating ability and high electron transport properties of diphenanthroline ligands, m-dPhen-based CAN system remains the high electron injection efficiency within a wide processing window of thicknesses (between 5 nm and 30 nm) and doping concentrations (between 5 wt% and 20 wt%), thus providing good manufacturing feasibility. Since the LUMOs levels of prevailing electron transport materials in organic solar cells (OSC) were mostly in the range from -4 to -3 eV⁹, The CAN-modified electrodes with low work function about 3.0 eV can also facilitate electron extraction in view of energy alignment. In addition, 2,9-dimethyl-4,7-diphenyl-1,10-phenanthroline (BCP):Ag complex have been successfully employed to reduce the electron extraction barrier between a C₆₀ electron transport layer and indium-zinc oxide top electrode in high-performance perovskite solar cells³⁶. Thus, CAN technique possessing high WF tunability, good thermal stability and wide processing window is anticipated to be applicable to the cathode modification in OLEDs, OSCs and other optoelectronic devices.”

36. Ying, Z. et al. Charge-transfer induced multifunctional BCP:Ag complexes for semi-transparent perovskite solar cells with a record fill factor of 80.1%. *J. Mater. Chem. A* **9**, 12009-12018 (2021)

Comment 8: The Figs. 1c, 1d, 4a and 5b are not clear. Figure legends should be added

in Fig. 4c.

Responses: Thanks for your valuable comment. We have modified the figures in the revised manuscript and added the figure legends in Fig. 4c.

Reviewer #2

General comment: This manuscript by Duan et al. demonstrated that efficient and stable electron injection can be achieved by incorporating novel diphenanthroline type ligands for in situ coordination-activated n-doping (CAN) techniques. The designed meta-linkage molecule (m-dPhen) shows high electron affinity with deep LUMO level (~ 3.0 eV) and strong nucleophilic quality by forming tetrahedrally coordinated double-helical structure, thus enabling high electron injection efficiency. The material design strategy of meta-linked diphenanthroline ligands proposed here successfully circumvents the limitations of the mono-Phen type ligands in previous reports of the same group as well as others, which are quite impressive and rigorous. And the results and discussions will surely impact future studies on this advanced n-doping technology. More importantly, the resulting deep-blue OLEDs shows high CCE of 237 cd/A and super-long lifetime with T95 of 104.1 h at 5000 cd/m², which are obviously the one of best results ever reported for deep-blue OLEDs. Overall, the manuscript presented an impressive advance for n-doping technique, which should also interest other organic electronic devices besides OLEDs. This manuscript is well written and topics are meaningful. Therefore, I think the paper could be accepted after further improvement.

Author reply: Thanks for your valuable comments and suggestions. Changes are highlighted by red-characters in the revised manuscript.

Comment 1: In the manuscript, the meta-linkage diphenanthroline type ligands are superior to mono-type Phen ligands for CAN strategies, an in-depth comparison and explanation should be added to present the advantages of proposed designing strategies.

Response: Thank you very much for your valuable comments and suggestions. We supposed that the advantages of meta-linkage diphenanthroline ligands could be summarized as follows:

Firstly, as plotted in Fig. 3e, the electron injection efficiency of meta-linkage diphenanthroline type ligands (m-dPhen) is higher than that of mono-type Phen ligands (p-MeO-Phen). Since the trade-off between nucleophilic quality and electron affinity has become the bottleneck for further improving the electron injection efficiency of mono-type Phen ligands, the molecular designing of meta-linkage diphenanthroline allows high electron affinity and strong chelating ability simultaneously, thus achieving higher electron injection for CAN strategies.

Secondly, due to the strong chelating ability and high electron transport properties of diphenanthroline ligands, m-dPhen-based CAN system achieves high electron injection efficiencies about 50% within a wide processing window of thicknesses (at least between 5 nm and 30 nm) and doping concentrations (at least between 5 wt% and 20 wt%), thus providing manufacturing feasibility and widespread application of such technology.

Thirdly, the application of typical mono-type Phen ligands such as bathophenanthroline (Bphen) is usually hampered by its low glass-transition temperature (T_g) of 66 °C (*Adv. Funct. Mater.* 2014, 24, 6540). However, m-dPPhen and m-dPhen possess high glass-transition temperatures over 130 °C (Supplementary Table 1), which are favorable for improving the thermal stability of resulting devices.

Based on these, the comparison and explanation about the advantages for meta-linkage diphenanthroline type ligands are added in the revised manuscript as follows:

“By decoupling nucleophilic quality and electron affinity, the meta-linked diphenanthroline-type ligand, m-dPhen, outperformed typical mono-Phen ligands of p-MeO-Phen for electron injection. As a result of the strong chelating ability and high electron transport properties of diphenanthroline ligands, m-dPhen-based CAN system remains the high electron injection efficiency within a wide processing window of thicknesses (between 5 nm and 30 nm) and doping concentrations (between 5 wt% and 20 wt%), thus providing good manufacturing feasibility. Since the LUMOs levels of prevailing electron transport materials in organic solar cells (OSC) were mostly in the range from -4 to -3 eV⁹, The CAN-modified electrodes with low work function about 3.0 eV can also facilitate electron extraction in view of energy alignment. In addition, 2,9-dimethyl-4,7-diphenyl-1,10-phenanthroline (BCP):Ag complex have been successfully employed to reduce the electron extraction barrier between a C₆₀ electron transport layer and indium-zinc oxide top electrode in high-performance perovskite solar cells³⁶. Thus, CAN technique possessing high WF tunability, good thermal stability and wide processing window is anticipated to be applicable to the cathode modification in OLEDs, OSCs and other optoelectronic devices.”

Comment 2: The authors confirmed the validity and compatibility of CAN in different device structures in supporting information. Apart from current density-voltage-brightness characteristics, the EL spectra of each OLEDs should be given to confirm the cavity effect.

Response: Thanks for your suggestion. The EL spectra of each OLEDs using CAN strategies were provided as below. At a brightness about 1000 cd m⁻², normalized EL spectra of OLEDs based on different EILs remain almost the same. This results can be interpreted as the thin thickness (5 nm) and high optical transmittance in the wavelength range of 400–700 nm for CAN-type EILs.

Supplementary Figure 22. (a) Deep-blue OLEDs using PAH-type α,β -ADN with LUMO of -2.9 eV as electron transporting materials. (b) Normalized EL spectra at a brightness about 1000 cd m^{-2} . (c) The current density–voltage of OLED devices with ETM of α,β -ADN and different EILs. (d) The brightness–voltage of OLED devices with ETM of α,β -ADN and different EILs. The device structure is ITO/HATCN (5 nm)/NPB (30 nm)/BCzPh (10 nm)/ α,β -ADN:*t*-DABNA (30 nm, 2 wt%)/CzPhPy (10 nm)/ α,β -ADN (20 nm)/EIL/Al. For the device implementing nitrogen-free α,β -ADN as ETMs, electroluminescence performances of OLEDs based on m-dPhen:Ag is significantly better than those with LiF, which agrees well with the electron injection efficiencies of m-dPhen:Ag and LiF of EODs in Supplementary Fig. 14.

Supplementary Figure 23. (a) Deep-blue OLEDs using silver with high work function of 4.5 eV as cathode. (b) Normalized EL spectra at a brightness about 1000 cd m⁻². (c) The current density–voltage of OLED devices with cathode of Ag and different EILs. (d) The brightness–voltage of OLED devices with cathode of Ag and different EILs. The device structure is ITO/HATCN (5 nm)/NPB (30 nm)/BCzPh (10 nm)/ α,β -ADN:*t*-DABNA (30 nm, 2 wt%)/CzPhPy (10 nm)/DPPyA (20 nm)/EIL/Ag. As plotted in Supplementary Fig. 23, the injection properties of Liq and LiF with silver cathodes are inferior to m-dPhen:Ag. This may be result from the insufficient coordination reaction at the interface of DPPyA/lithium compound/Ag. Differently, the Ag cathode modified by 5 nm of m-dPhen:Ag (10 wt%) shows a low work function about 3.5 eV, thus enables efficient electron injection.

Supplementary Figure 24. (a) Inverted green OLEDs with ITO as cathodes. (b) Normalized EL spectra at a brightness about 1000 cd m^{-2} . (c) The current density–voltage of inverted OLED devices with different EILs. (d) The brightness–voltage of inverted OLED devices with different EILs. The device structure is ITO/LiF(1 nm), m-dPhen:Ag (10 nm, 10 wt%), m-dPhen:Cs (10 nm, 5 wt%) or Cs (1 nm)/m-dPPhen (20 nm)/DIC-TRZ:Ir(ppy)₃ (25 nm, 7 wt%)/TCTA (10 nm)/NPB (30 nm)/HATCN (10 nm)/Al (150 nm). The results indicates that m-dPhen:Ag can also function as efficient EIL for ITO cathodes in inverted OLEDs.

Comment 3: The items and figure legends in figures 1, 4 and 5 should be modified to give clear information.

Response: Thanks for your valuable comment. We have carefully modified the figures 1, 4, and 5 to ensure that the information is clear in the revised manuscript.

Comment 4: The device performances of deep-blue OLEDs based on three diphenanthroline ligands should be summarized in Table.

Response: Thanks for your valuable comment. We have added Supplementary Table 2 to clearly summarize the device performances of deep-blue OLEDs based on three diphenanthroline ligands as follows:

Supplementary Table 2. Summary of the performances of deep-blue OLEDs based on

EILs consisting of Ag-doped diphenanthroline ligands.

EIL	Voltage ^{a)} (V)	EQE _{max/1000/5000} ^{b)} (%)	PE _{max/1000/5000} ^{b)} (lm W ⁻¹)	CIE (x, y)	Lifetime LT90 ^{c)} (h)
p-dPPhen:Ag	4.8	5.3/4.6/5.1	2.2/2.2/2.0	(0.132, 0.085)	31.2
m-dPPhen:Ag	4.4	7.8/7.6/6.9	5.1/4.1/2.9	(0.132, 0.084)	74.5
m-dPhen:Ag	4.2	10.3/10.3/9.5	6.6/5.8/4.4	(0.133, 0.085)	79.2

^{a)} Values at 1000 cd m⁻²; ^{b)} Maximum, at 1000 cd m⁻², at 5000 cd m⁻²; ^{c)} Lifetime up to 90% of initial brightness at 2000 cd m⁻².

Comment 5: The authors should give a brief explanation why the performances of deep-blue OLED with m-dPhen and Yb are compared by CCE.

Response: Thanks for your valuable comment. According to the luminous efficiency function, values of the $V(\lambda)$ function are relatively low in the blue region of the spectrum (around 460 nm). Therefore, the current efficiency of devices strongly depends on the color coordinate, especially for the deep-blue OLEDs with a narrow full-width at half-maximum (FWHM). The CCE value, which is calculated by dividing the current efficiency (cd A⁻¹) with CIEy color coordinate, was adopted to evaluate efficiency of deep-blue devices in previous reports (*Nat. Photon.* 2021, 15, 208; *Adv. Optical Mater.* 2021, 9, 2100203; *Org. Electron.* 2021, 95, 1106197). Accordingly, we compare the performances of deep-blue OLED with m-dPhen and Yb by CCE values. In the revised manuscript, the following discussion and references has been added in the revised manuscript:

“Since the current efficiency of deep-blue OLEDs with a narrow FWHM strongly depends on the color coordinate, the CCE value, which is calculated by dividing the current efficiency (cd A⁻¹) with CIEy color coordinate, was adopted to evaluate efficiency of deep-blue devices in previous reports⁴⁸⁻⁵².”

48. Ci, Z. et al. Fabrication of highly efficient blue top-emission organic light-emitting diodes on different reflective electrodes. *Org. Electron.* **95**, 106197 (2021).
49. Chung, J. W. et al. Over 30 000 h Device Lifetime in Deep Blue Organic Light-Emitting Diodes with y Color Coordinate of 0.086 and Current Efficiency of 37.0 cd A⁻¹. *Adv. Opt. Mater.* **9**, 2100203 (2021).
50. Cui, L. S. et al. Long-lived efficient delayed fluorescence organic light-emitting diodes using n-type hosts. *Nat. Commun.* **8**, 2250 (2017).
51. Chan, C. Y. et al. Stable pure-blue hyperfluorescence organic light-emitting

diodes with high-efficiency and narrow emission. *Nat. Photon.* **15**, 203-207 (2021).

52. Jeon, S. O. et al. High-efficiency, long-lifetime deep-blue organic light-emitting diodes. *Nat. Photon.* **15**, 208-215 (2021).

Reviewer #3

General comment: This manuscript employed the CAN strategy depending on metal-linked diphenanthroline-type ligands, co-evaporated with Silver, as n-type dopants to enhance electron injection efficiencies and thus achievement of high-efficiency deep-blue OLED with a high current efficiency (calibrated by y-color coordinate (0.0045) of 237 cdA-1 and superb LT95 of 104.1 hours at 5000 cdm-2). However, to improve the current manuscript, I believe the following comments should be addressed:

Author reply: We are grateful to your valuable comments and suggestions, which are helpful to improve the quality of our manuscript. We have revised the manuscript and the changes are highlighted by red-characters. Detailed responses are listed below.

Comment 1: The authors claim that the high electron injection efficiency is independent on the n-type dopant concentration of Ag. In Supplementary Figure 11(a), the authors compare the varied concentrations of 5, 10, 20 wt% to 0 wt% and the results displayed significantly enhanced current densities over the 0 wt% device. However, low dopant concentrations like 1 wt% or 3 wt% should be added to further investigate electron injection efficiencies with no impact placed by such low dopant concentrations when considering n-type doping as electrical doping. In addition, the data for the 0 wt% device should be added to Supplementary Figure 11(b).

Response: Thanks for your constructive suggestions. Given that the optimal Ag-doping concentration of mono-Phen ligands was as high as 20 wt% (*Nat. Commun.* 2019, 10, 866), we firstly compared the varied concentrations of 5, 10, 20 wt% to 0 wt% of diphenanthroline ligands in the initial manuscript to verify whether diphenanthroline ligands could break breaking the mutual exclusion between high nucleophilic quality and high electron affinity of mono-Phen ligands that limits the electron injection efficiency of CAN.

Following your comments, the electron-only device (EODs) with EILs of m-dPhen:Ag ($x = 0.5, 1$ and 3 wt%, 5 nm) were fabricated to further study the electron injection efficiencies at low dopant concentrations. The current density (J)-voltage (V) characteristics of EODs at various dopant concentrations (between 0 wt% and 20 wt%) and electron injection efficiencies at 100 kV cm⁻¹ were summarized as follows (Supplementary Fig. 12). As plotted in Supplementary Fig. 12b, with increasing dopant concentration, the electron injection efficiency of m-dPhen:Ag EIL at 100 kV cm⁻¹ gradually increased by two orders of magnitude from 0.24% with neat m-dPhen to 46% with 5 wt% (≈ 0.5 vol%) Ag doping. Because the coordination reaction between Ag and ligands is critical to the n-doping effect for CAN EILs, increasing dopant concentration were favorable to enhance in situ coordination reaction and thus larger work function downshift of cathodes modified by EILs. Since the injected currents of EODs depends exponentially on the electron injection barrier between work function of cathode and LUMO of electron transport materials, it can be anticipated that the electron injection efficiencies would show strong correlation with the dopant concentrations. Additionally, these results are in accordance with previous report that the shifts of work function were positively related to the dopant concentration for electrical p-type and n-type doping (*Nat. Mater.* 2019, 18, 149; *Light-Sci. Appl.* 2015,

4, e273; *Adv. Mater.* 2017, 29, 1701641). Given that UPS data in Supplementary Fig. 15 and the fact that silver-doping concentration of 5 wt% (0.5 vol%) was comparable to typical lithium-doping concentration of 5 wt% (*J. Mater. Chem. C* 2017, 5, 9911), the electron injection enhancement was thereof ascribed to the electrical n-type doping of silver dopants. With careful consideration, we modified the discussion about correlation between doping concentration and injection efficiency as follows:

“The influence of Ag-doping concentrations on injection performances was further studied. As shown in Supplementary Fig. 12b, the $\eta_{\text{injection}}$ of m-dPhen:Ag EIL at 100 kV cm^{-1} increases at first and then saturates at higher doping concentrations, which can be ascribed to the electrical n-type doping characters. It was intriguingly to note that $\eta_{\text{injection}}$ increases by two orders of magnitude from 0.24% with neat m-dPhen to 46% with 5wt% Ag-doping. The high electron injection efficiency at low doping concentration of 5 wt% (0.5 vol%) can also reflect strong coordination ability of m-dPhen, which is preferred in terms of optical transparency and fabrication robustness.”

“As a result of the strong chelating ability and high electron transport properties of diphenanthroline ligands, m-dPhen-based CAN system remains the high electron injection efficiency within a wide processing window of thicknesses (between 5 nm and 30 nm) and doping concentrations (between 5 wt% and 20 wt%), thus providing good manufacturing feasibility.”

“By optimizing the ligand structure, EILs with a wide processing window of thickness and doping concentrations were obtained, which were also applicable for a variety of ETMs, electrodes and device structures with injection efficiencies outperforming those of LiF and Liq.”

Supplementary Figure 12. (a) J-V characteristics of EODs with structures of ITO/Bphen:Cs₂CO₃ (10 nm, 10 wt%)/DPPyA (100 nm)/m-dPhen:Ag (0, 0.5, 1, 3, 5, 10 and 20 wt%, 5 nm)/Al (150 nm). (b) Injection efficiency of EODs based on different EILs of m-dPhen:Ag (0, 0.5, 1, 3, 5, 10 and 20 wt%, 5 nm). The electron injection efficiency of m-dPhen:Ag at lower concentration of 5 wt% (0.5 vol%) is very close to those at 10 wt% (1 vol%) and 20 wt% (2 vol%), which can be interpreted by the strong

coordination ability of m-dPhen.

Comment 2: The authors built one set of EODs employing different EILs, including Ag-doped diphenanthroline ligands, commonly-used LiF and Liq with the same device structure (Figure 3c). However, the thickness of the EILs varied from 5 nm to 1 nm before investigation of electron injection efficiencies independent on the EIL thickness. Logically, the same thickness of EILs should be maintained (1 nm) for a solid “apples to apples” comparison while focusing on one variable at a time. Despite the achievement of an excellent result about the high electron injection efficiency independent on thickness of EILs in Figure 3e, the design of the EIL thickness has nonetheless remained different regardless of the consideration of the typical thickness range from 0.5 nm to 2 nm for EILs such as LiF and Liq.

Response: Thanks for your constructive comments. In this part, the EODs employing Ag-doped diphenanthroline ligands and lithium compounds were fabricated to compare maximum electron injection efficiencies of different EILs. Since the optimal thickness for different EILs were closely related to the injection mechanism, the thicknesses of different EILs were set as the previously reported optimal values in the initial attempt (*Nat. Commun.* 2021, 12, 2706; *Nat. Commun.* 2019, 10, 866). So lithium compounds with thickness of 1 nm and Ag-doped diphenanthroline ligands of 5 nm were employed as EILs to compare the injection properties. Following your suggestions, we have further measured *J-V* characteristics of EODs with various EILs at different thicknesses to verify the optimal thickness of these EILs. The results are summarized as below:

Supplementary Figure 11. (a) The J-V characteristics of EODs with LiF at thicknesses of 0.5, 1, 1.5 and 2 nm. (b) The J-V characteristics of EODs with Liq at thicknesses of 0.5, 1, 1.5 and 2 nm. (c) The J-V characteristics of EODs with p-dPPhen:Ag at thicknesses of 1, 3, 5 and 10 nm. (d) The J-V characteristics of EODs with m-dPPhen:Ag at thicknesses of 1, 3, 5 and 10 nm. (e) The J-V characteristics of EODs with m-dPhen:Ag at thicknesses of 1, 3, 5 and 10 nm. (f) The Injection efficiency-thickness of different EILs.

As shown in Supplementary Fig. 11, the optimal thickness for EILs consisting of lithium compounds is about 1 nm. Tang et al. demonstrated that the interfacial reaction ($3\text{LiF} + \text{Al} + 3\text{ETM} \rightarrow \text{AlF}_3 + 3\text{Li}^+ \text{ETM}^-$) and the formation of n-doped ETM anions was critical to the improved electron injection for lithium compounds (*J. Appl. Phys.* 2001, 89, 2756). Therefore, the optimal thickness of typical LiF and Liq were expected to be about 1 nm so that thermally hot Al atom can react with isolated clusters of lithium compound and ETMs. Because the electron injection enhancement of Ag-doped EILs

was attributed to the work function downshifts of cathode modification, the formation homogeneous surface with high coverage is essential to enable efficient cathode modification. In our previous study (*Nat. Commun.* 2019, 10, 866), it was found that co-evaporation of Ag and Phen-ligands can form thin film (5 nm) with low root-mean-square (rms) of 1.1 nm. Therefore, the thickness of n-doped phenanthroline-type ETMs should be higher than 5 nm to achieve maximum injection efficiency. Based on these, lithium compounds with thickness of 1 nm and Ag-doped diphenanthroline ligands of 5 nm were employed in Figs. 3c and 3d to compare the injection properties. Given the length restrictions of the manuscript, the analysis about injection efficiency-thickness of different EILs were added in the Supplementary Fig. 11. The manuscript has been revised as follows:

“EODs employing different EILs including Ag-doped diphenanthroline ligands, and conventional ones such as LiF³⁴ and Liq³⁵ were examined with the structure of ITO/Bphen:Cs₂CO₃ (10 nm, 20 wt%)/DPPyA (100 nm)/EIL/Al (150 nm). To reasonably evaluate the electron injection properties of different EILs, we have further measured the *J-V* characteristics of EODs with various EILs at different thickness to verify their optimal thicknesses (Supplementary Fig. 11), thus determining their maximum electron injection efficiencies in the EODs. Accordingly, lithium compounds with thickness of 1 nm and Ag-doped diphenanthroline ligands of 5 nm were employed for comparison. The *J-V* characteristics of EODs employing different EILs with optimal thickness are summarized in (Fig. 3c), with m-dPhen:Ag (5 nm, 10 wt%) being the best.”

Supplementary Figure 11. (a) The J-V characteristics of EODs with LiF at thicknesses of 0.5, 1, 1.5 and 2 nm. (b) The J-V characteristics of EODs with Liq at thicknesses of 0.5, 1, 1.5 and 2 nm. (c) The J-V characteristics of EODs with p-dPPhen:Ag at thicknesses of 1, 3, 5 and 10 nm. (d) The J-V characteristics of EODs with m-dPPhen:Ag at thicknesses of 1, 3, 5 and 10 nm. (e) The J-V characteristics of EODs with m-dPhen:Ag at thicknesses of 1, 3, 5 and 10 nm. (f) The Injection efficiency-thickness of different EILs. The optimal thickness of typical LiF and Liq was set as 1 nm so that thermally hot Al atom can react with isolated clusters of lithium compound and ETMs. Because the electron injection enhancement of Ag-doped EILs was attributed to the work function downshifts of cathode modification, the formation homogeneous surface with high coverage is essential to enable efficient cathode modification. Based on these, lithium compounds with thickness of 1 nm and Ag-doped diphenanthroline ligands of 5 nm were employed in Figs. 3c and 3d to compare the injection properties.

Comment 3: The UPS characterizations prove that the work functions of different cathode electrodes have been reduced due to the deposition of a 5-nm-thick m-dphen:Ag (10 wt%) n-dopant, which has modified the interface of cathodes. Forrest et al. had reported that the smooth and homogeneous surface of nano-scale films can be

achieved via thermal evaporation, a typical deposition technique for OLEDs fabrication (Chem. Rev. 1997, 97, 1793-1896). Moreover, to our knowledge, sensitivity of UPS source is limited within a few nanometers less than 5 nm. Theoretically, only the Fermi level can be characterized for the shallow surface of 5 nm-thick films rather than the work function of the bulk cathode material. However, the results definitely demonstrate the reduced work function of individual cathodes upon top-deposition of a 5-nm-thick m-dphen:Ag on the cathode electrodes. Nevertheless, I would like the authors to please provide the result of calibrated thickness of the co-evaporated EILs and measure their intrinsic Fermi levels by depositing thick (> 20 nm) films using UPS to further support their conclusions.

Response: Thanks for your constructive comments. We tried to measure the calibrated thickness of thin EIL with assumed thickness of 5 nm in situ by atomic force microscope (AFM), and found that it was difficult to distinguish signals of EIL film from the signals of substrate in the nano-scale. For OLEDs fabrication, the film thickness during thermal evaporation was usually monitored by quartz crystal microbalance (QCM), which was a widely-used and extremely sensitive method. When using the same evaporation source, the frequency shift (Δf) of the quartz crystal sensor was linear to the mass load (Δm) of deposited films on the substrate (*Appl. Phys. Lett.* 2007, 90, 012119). Considering that the mass load (Δm) of deposited films are also linear to the film thickness (Δd) when using the same mask, the frequency shift (Δf) should be proportional to the film thickness (Δd). Based on this, we fabricated thick Ag-doped EILs whose frequency shift (Δf) was 20 times as large as that of “5 nm” films. As the following Figure showed, the thickness of thick Ag-doped EILs was determined to be 107.9 nm by dektak profilometer. So the calibrated thickness of the m-dPhen:Ag was estimate to be about 5.4 nm.

Following your comments, we used UPS to measure the energy level alignment of aluminum cathodes modified by Ag-doped EILs with thickness of 5, 10, 20, 30 nm. As plotted in Supplementary Fig. 13, the line shapes, HOMO onsets and work functions with EIL thickness ranging from 5 to 30 nm are similar for the each diphenanthroline ligand, but different from the signals of Al cathodes. The phenomenon was in accordance with previous report that the work function downshifts owing to n-doped phenanthroline ETMs nearly saturated with the layer thickness increasing from 5 nm to 20 nm (*Org. Electron.* 2012, 13, 2346). This could be attributed to the formation of

smooth and homogeneous surface for layer about 5 nm and the strong n-doping effect on cathode modification (*Nat. Commun.* 2019, 10, 866). Accordingly, the manuscript has been revised as follows:

“Moreover, it is found that a 5-nm-thick m-dPhen:Ag (10 wt%) would lead to WF reduction for ITO from 4.6 to 3.1 eV and Ag from 4.5 to 3.5 eV, respectively, as illustrated in Fig. 3f. Since the thermal evaporation enables the formation of smooth and homogeneous thin layers in nano-scale³⁷, WF downshifts of Al cathodes modified by Ag-doped EILs nearly saturate when the thickness exceeds 5 nm^{22,38} (Supplementary Fig. 13). Therefore, it is inferred that m-dPhen:Ag (5 nm, 10 wt%) can act as an efficient and universal EIL for optoelectronic devices with different electrodes.”

37. Forrest, S. R. Ultrathin Organic Films Grown by Organic Molecular Beam Deposition and Related Techniques. *Chem. Rev.* **97**, 1793-1896 (1997).
38. Lee, S. et al. Determination of the interface energy level alignment of a doped organic hetero-junction using capacitance–voltage measurements. *Org. Electron.* **13**, 2346-2351 (2012).

Supplementary Figure 13. (a) UPS data of aluminum cathode modified by p-dPPhen:Ag with thickness of 0, 5, 10, 20 and 30 nm. (b) UPS data of aluminum cathode modified by m-dPPhen:Ag with thickness of 0, 5, 10, 20 and 30 nm. (c) UPS data of aluminum cathode modified by m-dPhen:Ag with thickness of 0, 5, 10, 20 and 30 nm.

(c) The work functions of aluminum cathode modified by different Ag-doped EILs with thickness of 5, 10, 20 and 30 nm.

Comment 4: The calculated IEs for Ag(m-dphen) and Ag((m-dphen)₂) denoted as 3.45 eV and 3.06 eV in the main manuscript text, which are different from the values displayed in Figure 3b.

Response: Thanks for your comment and suggestion. We apologize for the mistakes. After rechecking the results of theoretical calculation, the manuscript was revised as follows:

“Interestingly, Ag(m-dPhen) is found to possess the lowest calculated IE of 3.47 eV among all organometallic complexes with one dPhen ligand. The reason is assigned to the unique U-shaped structure of m-dPhen, inducing synergetic effect between the two Phen units to enlarge the ESP. The IEs of tetrahedrally coordinated metal complexes (AgL₂) were also calculated to be 3.11, 3.10 and 3.05 eV for p-dPPhen, m-dPPhen and m-dPhen, respectively.”

Comment 5: Minor: A few syntax errors and typos are found by this reviewer. Examples:

- Abstract line 9: is achieved;
- Results line 97: Different from;
- Results line 145: (UPS) are.

To be accepted by Nature Communications, the reviewer believes that any similar errors or typos should be addressed.

Response: Thanks for your suggestion. We are sorry for the syntax errors and typos in the manuscript. We have carefully checked and revised the manuscript.

In Abstract line 9, “is achieved” was corrected as “is developed”;

In Results line 97, “Different from typical EILs of lithium compounds, m-dPhen:Ag (5 nm, 10 wt%) allows efficient electron injection even for PAH-type ETMs (Supplementary Fig. 20b).” was corrected as “It is worthy to note that EILs of m-dPhen:Ag allow efficient electron injection even for PAH-type ETMs (Supplementary Fig. 14b), which is different from typical EILs of lithium compounds”;

In Results line 145, “ultraviolet photoelectron spectroscopy (UPS) are conducted to verify their roles in cathode WFs tunability.” was corrected as “ultraviolet photoelectron spectroscopy (UPS) measurement was conducted to verify their roles in cathode WFs tunability.”;

Reviewer #4

General comment: This manuscript concerns itself with coordination-activated doping using organic ligands. The authors describe a fundamental problem with this approach, namely that the ligand must be nucleophilic to form the bond, but should also have a high electron affinity for efficient n-doping. A chelating ligand is shown to overcome this issue, and highly efficient OLEDs are manufactured using this electron transport layer.

Author reply: We are grateful to your valuable comments and suggestions, which will help us improve the quality of the manuscript. Changes are highlighted by red-characters in the revised manuscript.

Comment 1: The synthesized materials, layers and devices are carefully characterized by a very large number of techniques (maybe even a confusingly large number of techniques, but if it fits within length restrictions of the manuscript that is OK). The methodology appears sound.

Response: Thanks for your constructive comments. With careful consideration, some results were plotted in the Supplementary Information and the manuscript was revised to satisfy length restrictions.

Comment 2: The motivation for this work is in a large part taken from a desire to compete with current industry standards for OLED efficiency. For example, it is stated that “The OLEDs industry with multi-billion outputs today still relies on highly reactive metals to ensure desirable device performances despite of hazardous reactivity and diffusivity.” No citation is attached to this statement, and moreover, it is hard to know that because of likely trade-secrets. The potential impact of this work is therefore difficult to judge for me. There is an informative table (Table 2) in the supplementary materials. From that table it appears that the progress made here is not particularly significant in terms of efficiency. I would like the authors to comment on this point.

Response: Thanks for your constructive comments and suggestions. We were sorry that we didn't make clear the highlights of this work in the initial manuscript. To the best of our knowledge, lithium (Li) with low work function (2.9 eV) was the most widely used n-dopants for the OLED industry, especially for the charge generation layer (CGL) in tandem OLEDs (*Adv. Mater.* 2016, 28, 10381). In these Li-based CGLs, the lithium-doped layers were employed to enable efficient electron injection into the electron transport layers, thus achieving high efficiency and low driving voltage. LG Display Co. focused on the commercial application of tandem white OLEDs in large-sized and high resolution OLED TV (*SID Symp. Dig. Tech. Pap.* 2012, 43, 279; *SID Symp. Dig. Tech. Pap.* 2016, 47, 707; *SID Symp. Dig. Tech. Pap.* 2017, 48, 1), where lithium was used as n-dopants in the n-type CGL (*SID Symp. Dig. Tech. Pap.* 2011, 19, 190). Recently, Bae et al. introduced a thin CGL with structure of lithium-doped electron transport layers (5 wt%, 5 nm)/HATCN (10 nm)/hole transport layers to achieve high performance white OLEDs and discussed the potential application as microdisplays in Virtual reality (VR) and augmented reality (AR) (*ACS Appl. Electron.*

Mater. 2021, 3, 3240).

However, lithium is very sensitive to the moisture, oxygen and even nitrogen in the ambient atmosphere, so the highly reactive lithium have to be carefully stored and handled, which hinders the manufacturing feasibility and widespread application. Additionally, Forrest et al. reported that metallic Li would easily diffuse through the organic layers, which would lead to severe exciton quenching in the emitting layers and resulting in device degradation (*J. Appl. Phys.* 2001, 89, 4986). Slyke et al. developed a novel Li-free P-N connector for tandem OLEDs, and confirmed that this Li-free connector helped in eliminating dopants diffusion and improving the lifetimes of tandem white OLEDs (*SID Symp. Dig. Tech. Pap.* 2009, 40, 499). Recently, tremendous efforts have been devoted to develop air-stable n-dopants for OLEDs (*Nat. Mater.* 2017, 16, 1209; *Adv. Mater.* 2019, 31, 1904201). In this manuscript, it was found that electron injection layers using air-stable silver as n-dopants for diphenanthroline ligands enables high-efficiency and long-lifetime OLEDs. So we aimed to provide an alternative n-doping technique, which may pave the way towards replacing the reactive metals (Li) in OLEDs industry.

For the device application, by virtue of deep-blue emission, small efficiency roll-off, low operation voltage, and superior operational stability, TTA-OLEDs are still popular for commercial application. (*Adv. Mater.* 2021, DOI: 10.1002/adma.202100704; *Org. Electron.* 2019, 75, 105377). Therefore, we evaluate the overall performances of EILs based on CAN technique to evidence the potential application in OLED industry. Reviewer #1 said “the corresponding deep-blue fluorescent OLEDs achieved impressive CCE of 237 cd A⁻¹ at 1000 cd m⁻² and record-high T95 > 100 h at 5000 cd m⁻², satisfying the requirement for industrial application.” Reviewer #2 also commented “the resulting deep-blue OLEDs shows high CCE of 237 cd/A and super-long lifetime with T95 of 104.1 h at 5000 cd/m², which are obviously the one of best results ever reported for deep-blue OLEDs.” Combing with Supplementary Table 4 and Table 5, we supposed that the OLEDs with EQE_{max} of 10.3% and devices with CCE_{max} of 237 cd A⁻¹ in this manuscript was among the most efficient deep-blue TTA-OLEDs.

To further verify the widespread applicability of proposed n-doping strategies, we constructed a set of OLEDs by replacing the emitting layer with phosphorescent dopants and thermally activated delayed fluorescent (TADF) materials. As shown in Supplementary Fig. 20, high EQEs exceeding 30% and improved power efficiency were attained for blue, green and red OLEDs due to the enhanced electron injection, which suggested that incorporating efficient electron injection layers based on in situ coordination-activated n-doping can also contributes to high-performance OLEDs.

In order to avoid misunderstanding and to be more accurate, we changed our expression as follows in the revised manuscript and added new citations about the application of lithium in commercialized products.

“Despite of hazardous reactivity and diffusivity, the highly reactive metals were still widely employed today to facilitate efficient n-doping and ensure desirable device performances towards commercial application¹⁰⁻¹².”

10. Parthasarathy, G., Shen, C., Kahn, A. & Forrest, S. R. Lithium doping of semiconducting organic charge transport materials. *J. Appl. Phys.* **89**, 4986-4992 (2001).
11. Hatwar, T. K. et al. High-Performance Tandem White OLEDs Using a Li-Free “P-N” Connector. *SID Symp. Dig. Tech. Pap.* **40**, 499-502 (2009).
12. Bae, H. W., Kwon, Y. W., An, M., Kwon, J. H. & Lee, D. High-Color-Stability and Low-Driving-Voltage White Organic LightEmitting Diodes on Silicon with Interlayers of Thin Charge Generation Units for Microdisplay Applications. *ACS Appl. Electron. Mater.* **3**, 3240-3246 (2021)

“By replacing the emitting layer with phosphorescent dopants and thermally activated delayed fluorescent (TADF) materials, high EQEs exceeding 30% and improved power efficiency were attained for blue, green and red OLEDs due to the enhanced electron injection, which suggested that incorporating efficient electron injection layers based on CAN also contributes to high-performance OLEDs (Supplementary **Fig. 20** and Supplementary **Table. 3**).”

Supplementary Figure 20. (a) The devices structures of blue, green and red OLEDs based on different EILs of LiF (1 nm) or m-dPhen:Ag (5 nm). (b) The current density-voltage-luminance ($J-V-L$) and (c) the EQE-power efficiency-luminance curves of blue TADF OLEDs. (d) The $J-V-L$ and (e) the EQE-power efficiency-luminance curves of green TADF OLEDs. (f) The $J-V-L$ and (g) the EQE-power efficiency-luminance curves of red phosphorescent OLEDs.

Supplementary Table 3. Summary of the performances of blue, green and red OLEDs based on different EILs of LiF (1 nm) or m-dPhen:Ag (5 nm) in Supplementary Fig. 20

EIL	Emitting dopants	Voltage ^{a)} (V)	EQE _{max/1000/5000} ^{b)} (%)	PE _{max/1000/5000} ^{b)} (lm W ⁻¹)
LiF	TDBA-DI	5.6	32.8/22.1/13.8	46.0/18.8/8.6
m-dPhen:Ag	TDBA-DI	5.8	34.4/24.4/17.4	50.9/22.2/12.1
LiF	DACT-II	5.2	30.2/25.7/17.4	70.4/47.4/24.3
m-dPhen:Ag	DACT-II	5.0	30.4/25.6/17.9	73.8/49.6/26.9
LiF	Ir(mphmq) ₂ (tmd)	3.0	31.0/30.6/27.3	46.0/40.3/30.3
m-dPhen:Ag	Ir(mphmq) ₂ (tmd)	2.8	32.8/32.5/28.1	47.1/43.4/29.8

^{a)} Values at 1000 cd m⁻²; ^{b)} Maximum, at 1000 cd m⁻², at 5000 cd m⁻².

Comment 3: Regarding the characterization of the OLED devices, the authors state “The performances of EILs consisted of 10 wt% Ag-doped ligands (5 nm) were systematically evaluated in OLEDs with 1-nm LiF and 1-nm Liq as references.” I hope this means that LiF reference devices were compared to devices made from the materials proposed here, and that the latter devices did not contain LiF. Is this correct? The authors should state this more clearly. Otherwise LiF in one form would be compared to LiF in another form, and not to the pure materials proposed here.

Response: Thanks for your valuable comments. We were sorry for the ambiguous statement in the initial manuscript. As the reviewer pointed out, we would like to compare the devices made from the synthesized materials with the LiF reference devices herein. So we revised the description as follows:

“These vacuum-deposited CAN-type EILs possessed high transmittance above 95% in the whole visible region, suggesting that the Ag-doped diphenanthroline ligands could function as highly transparent charge injection layers. Consequently, the performances of these Ag-doped EILs (5 nm) were systematically evaluated in OLEDs. And reference devices based on EILs of 1-nm LiF and 1-nm Liq were also fabricated for comparison.”

Comment 4: Another question I have is why were TTA devices studied, wouldn't it have been more straightforward (and scientifically clearer) to study charge injection efficiency is direct charge recombination emitters, rather than the indirect TTA process?

Response: Thanks for your valuable comments. To straightforward study the charge injection efficiencies, we fabricated electron-only devices (EODs) with different electron injection layers and determined the charge injection efficiencies ($\eta_{\text{injection}}$) by equation of $\eta_{\text{injection}} = J_{\text{EOD}}/J_{\text{SCLC}}$, which have been proved effective for other charge injection layers. As shown in Figs. 3c and 3d, the m-dPhen:Ag showed highest injection

efficiency of 50% among three Ag-doped EILs and two lithium compounds. To evidence the applicability of these EILs on OLEDs, we employed TTA devices based on the following considerations. Firstly, given that TTA rate can be increased by increasing triplet exciton concentration, Kwon et al. demonstrated that the improvement of charge carrier injection and carrier balance in the EML could enhance charge recombination characteristics as well as triplet harvesting by TTA process, thus achieving high efficiency OLEDs (*Org. Electron.* 2019, 70, 1). As plotted in Supplementary Fig. 15, it was found that LiF-based reference devices suffered from insufficient electron injection and the electrons were the minority carriers for the emitting layers. Therefore, incorporating more efficient EIL of m-dPhen:Ag was reasonably beneficial to improving charge balance and boosting the performances of TTA devices.

Supplementary Figure 15. The current density-electrical field characteristics of hole-only half-devices (HOHDs) and electron-only half-devices (EOHDs). The device structure of HOHD (black) is ITO/HATCN (5 nm)/NPB (30 nm)/BCzPh (10 nm)/ α,β -ADN (30 nm)/Al (150 nm); The device structure of EOHDs are ITO/ α,β -ADN (30 nm)/CzPhPy (10 nm)/DPPyA (30 nm)/EIL (1 nm for LiF and Liq, 5 nm for Ag-doped p-dPPhen, m-dPPhen and m-dPhen)/Al (150 nm).

Secondly, owing to the high requirement for commercial application, the OLED industry still rely on TTA-OLEDs in blue device technology due to the advantages of deep-blue emission, small efficiency roll-off, low operation voltage, and superior operational stability (*Adv. Mater.* 2021, DOI: 10.1002/adma.202100704; *Org. Electron.* 2019, 75, 105377). Therefore, we employed deep-blue TTA devices with Ag-doped EILs to testify the potential application for OLED industry. In the revised manuscript, the correlation between electron injection efficiencies of EILs and TTA process were added as follows:

“Furthermore, the hole/electron currents of single carrier devices were characterized to study the charge balance of the devices based on different EILs. As shown in Supplementary Fig. 19, the LiF-based devices suffered from insufficient electron

injection and the electrons were the minority carriers in the emitting layers thereof. It has been demonstrated that enhanced charge carrier injection and improved carrier balance in the emitting layers could promote charge recombination characteristics as well as triplet harvesting by TTA process⁴². So incorporating a more efficient EIL of m-dPhen:Ag would be beneficial to the device performances.”

42. Bae, H. W. et al. Efficiency enhancement in fluorescent deep-blue OLEDs by boosting singlet exciton generation through triplet fusion and charge recombination rate. *Org. Electron.* **70**, 1-6 (2019).

After receiving the authors' responses I will make a recommendation regarding publication.

REVIEWER COMMENTS second round -

Reviewer #1 (Remarks to the Author):

I think that this revised manuscript is suitable for publication without further revision.

Reviewer #2 (Remarks to the Author):

I recommend this manuscript being accepted without changes.

Reviewer #4 (Remarks to the Author):

I have studied the responses to the reviewers and changes made to the manuscript. I am satisfied with those responses, and although some scientific aspects remain open to discussion, overall I now can recommend publication.